# MEASURING THE PREDICTIVE HETEROGENEITY

**Jiashuo Liu[†], Jiayun Wu[†], Renjie Pi[‡], Renzhe Xu[†], Xingxuan Zhang[†], Bo Li[†], Peng Cui[†*]**
[†]Tsinghua University, [‡]Hong Kong University of Science and Technology
{liujiashuo77, jiayun.wu.work}@gmail.com, rpi@connect.ust.hk
xrz199721@gmail.com, xingxuanzhang@hotmail.com
libo@sem.tsinghua.edu.cn, cuip@tsinghua.edu.cn

## ABSTRACT

As an intrinsic and fundamental property of big data, data heterogeneity exists in a variety of real-world applications, such as in agriculture, sociology, health care, etc. For machine learning algorithms, the ignorance of data heterogeneity will significantly hurt the generalization performance and the algorithmic fairness, since the prediction mechanisms among different sub-populations are likely to differ. In this work, we focus on the data heterogeneity that affects the prediction of machine learning models, and first formalize the *Predictive Heterogeneity*, which takes into account the model capacity and computational constraints. We prove that it can be reliably estimated from finite data with PAC bounds even in high dimensions. Additionally, we propose the Information Maximization (IM) algorithm, a bi-level optimization algorithm, to explore the predictive heterogeneity of data. Empirically, the explored predictive heterogeneity provides insights for sub-population divisions in agriculture, sociology, and object recognition, and leveraging such heterogeneity benefits the out-of-distribution generalization performance.

## 1 INTRODUCTION

Big data bring great opportunities to modern society and promote the development of machine learning, facilitating human life within a wide variety of areas, such as the digital economy, healthcare, scientific discoveries. Along with the progress, the intrinsic heterogeneity of big data introduces new challenges to machine learning systems and data scientists (Fan et al., 2014; He, 2017). In general, data heterogeneity, as a fundamental property of big data, refers to any diversity inside data, including the diversity of data sources, data generation mechanisms, sub-populations, data structures, etc. When not properly treated, data heterogeneity could bring pitfalls to machine learning systems, especially in high-stake applications, such as precision medicine, autonomous driving, and financial risk management (Dzobo et al., 2018; Breitenstein et al., 2020; Challen et al., 2019), leading to poor out-of-distribution generalization performances and some fairness issues. For example, in supervised learning tasks where machine learning models learn from data to predict the target variable with given covariates, when the whole dataset consists of multiple sub-populations with shifts or different prediction mechanisms, traditional machine learning algorithms will mainly focus on the majority but ignore the minority. It will hurt the generalization ability and compromise the algorithmic fairness, as is shown in (Kearns et al., 2018; Sagawa et al., 2019; Duchi & Namkoong, 2021). Another well-known example is Simpson's paradox, which brings false discoveries to the social research (Wagner, 1982; Hernán et al., 2011).

Despite its widespread existence, due to its complexity, data heterogeneity has not converged to a uniform formulation so far, and has different meanings among different fields. Li & Reynolds (1995) define the heterogeneity in *ecology* based on the system property and complexity or variability. Rosenbaum (2005) views the uncertainty of the potential outcome as unit heterogeneity in observational studies in *economics*. More recently, in machine learning, several works of *causal learning* (Peters et al., 2016; Arjovsky et al., 2019; Koyama & Yamaguchi, 2020; Liu et al., 2021; Creager et al., 2021) and *robust learning* (Sagawa et al., 2019; Liu et al., 2022) leverage heterogeneous data from multiple environments to improve the out-of-distribution generalization ability. However, previous works have not provided a precise definition or sound quantification. In this

---
[*]Corresponding Author.

work, from the perspective of *prediction power*, we propose the predictive heterogeneity, a *new type* of data heterogeneity.

From the machine learning perspective, the main concern is the possible negative effects of data heterogeneity on making predictions. Therefore, given the complexity of data heterogeneity, in this work, we focus on the data heterogeneity that affects the prediction of machine learning models, which could facilitate the building of machine learning systems, and we name it the ***predictive heterogeneity***. We raise the precise definition of predictive heterogeneity, which is quantified as the *maximal additional predictive information* that can be gained by dividing the whole data distribution into sub-populations. The new measure takes into account the model capacity and computational constraints, and can be reliably estimated from finite samples even in high dimensions with PAC bounds. We theoretically analyze its properties and examine it under *typical cases of data heterogeneity* (Fan et al., 2014). Additionally, we design the ***information maximization (IM)*** algorithm to empirically explore the predictive heterogeneity inside data. Empirically, we find the explored heterogeneity is explainable and it provides insights for sub-population divisions in many fields, including *agriculture, sociology, and object recognition*. And the explored sub-populations could be leveraged to enhance the out-of-distribution generalization performances of machine learning models, which is verified with both simulated and real-world data.

## 2 PRELIMINARIES ON MUTUAL INFORMATION AND PREDICTIVE $\mathcal{V}$-INFORMATION

In this section, we briefly introduce the mutual information and predictive $\mathcal{V}$-information (Xu et al., 2020) which are the preliminaries of our proposed predictive heterogeneity.

**Notations.** For a probability triple $(\mathbb{S}, \mathcal{F}, \mathbb{P})$, define random variables $X : \mathbb{S} \to \mathcal{X}$ and $Y : \mathbb{S} \to \mathcal{Y}$ where $\mathcal{X}$ is the covariate space and $\mathcal{Y}$ is the target space. Accordingly. $x \in \mathcal{X}$ denotes the covariates, and $y \in \mathcal{Y}$ denotes the target. Denote the set of random categorical variables as $\mathcal{C} = \{C : \mathbb{S} \to \mathbb{N}| \operatorname{supp}(C) \text{ is finite}\}$. Additionally, $\mathcal{P}(\mathcal{X}), \mathcal{P}(\mathcal{Y})$ denote the set of all probability measures over the Borel algebra on the spaces $\mathcal{X}, \mathcal{Y}$ respectively. $H(\cdot)$ denotes the Shannon entropy of a random variable, and $H(\cdot|\cdot)$ denotes the conditional entropy of two random variables.

In information theory, the mutual information of two random variables $X, Y$ measures the dependence between the two variables, which quantifies the reduction of entropy for one variable when observing the other:

$$\mathbb{I}(X; Y) = H(Y) - H(Y|X). \tag{1}$$

It is known that the mutual information is associated with the predictability of $Y$ (Cover Thomas & Thomas Joy, 1991). While the standard definition of mutual information unrealistically assumes the unbounded computational capacity of the predictor, rendering it hard to estimate especially in high dimensions. To mitigate this problem, Xu et al. (2020) raise the predictive $\mathcal{V}$-information under realistic computational constraints, where the predictor is only allowed to use models in the predictive family $\mathcal{V}$ to predict the target variable $Y$.

**Definition 1** (Predictive Family (Xu et al., 2020)). *Let* $\Omega = \{f : \mathcal{X} \cup \{\emptyset\} \to \mathcal{P}(\mathcal{Y})\}$. *We say that* $\mathcal{V} \subseteq \Omega$ *is a predictive family if it satisfies:*

$$\forall f \in \mathcal{V}, \ \forall P \in \operatorname{range}(f), \ \exists f' \in \mathcal{V}, \quad s.t. \ \forall x \in \mathcal{X}, f'[x] = P, f'[\emptyset] = P. \tag{2}$$

A predictive family contains all predictive models that are allowed to use, which forms computational or statistical constraints. The additional condition in Equation 2 means that the predictor can always ignore the input covariates ($x$) if it chooses to (only use $\emptyset$).

**Definition 2** (Predictive $\mathcal{V}$-information (Xu et al., 2020)). *Let* $X, Y$ *be two random variables taking values in* $\mathcal{X} \times \mathcal{Y}$ *and* $\mathcal{V}$ *be a predictive family. The predictive* $\mathcal{V}$*-information from* $X$ *to* $Y$ *is defined as:*

$$\mathbb{I}_{\mathcal{V}}(X \to Y) = H_{\mathcal{V}}(Y|\emptyset) - H_{\mathcal{V}}(Y|X), \tag{3}$$

*where* $H_{\mathcal{V}}(Y|\emptyset)$, $H_{\mathcal{V}}(Y|X)$ *are the predictive conditional* $\mathcal{V}$*-entropy defined as:*

$$H_{\mathcal{V}}(Y|X) = \inf_{f \in \mathcal{V}} \mathbb{E}_{x,y \sim X,Y}[-\log f[x](y)]. \tag{4}$$

$$H_{\mathcal{V}}(Y|\emptyset) = \inf_{f \in \mathcal{V}} \mathbb{E}_{y \sim Y}[-\log f[\emptyset](y)]. \tag{5}$$

*Notably that* $f \in \mathcal{V}$ *is a function* $\mathcal{X} \cup \{\emptyset\} \to \mathcal{P}(\mathcal{Y})$, *so* $f[x] \in \mathcal{P}(\mathcal{Y})$ *is a probability measure on* $\mathcal{Y}$, *and* $f[x](y) \in \mathbb{R}$ *is the density evaluated on* $y \in \mathcal{Y}$. $H_{\mathcal{V}}(Y|\emptyset)$ *is also denoted as* $H_{\mathcal{V}}(Y)$.

Compared with the mutual information, the predictive $\mathcal{V}$-information restricts the computational power and is much easier to estimate in high-dimensional cases. When the predictive family $\mathcal{V}$ contains all possible models, i.e. $\mathcal{V} = \Omega$, it is proved that $\mathbb{I}_\mathcal{V}(X \to Y) = \mathbb{I}(X; Y)$ (Xu et al., 2020).

## 3 PREDICTIVE HETEROGENEITY

In this paper, from the machine learning perspective, we quantify the data heterogeneity that affects decision making, named Predictive Heterogeneity, which is easy to integrate with machine learning algorithms and could help analyze big data and build more rational algorithms.

### 3.1 INTERACTION HETEROGENEITY

To formally define the predictive heterogeneity, we begin with the formulation of the interaction heterogeneity. The *interaction heterogeneity* is defined as:

**Definition 3** (Interaction Heterogeneity). *Let $X, Y$ be random variables taking values in $\mathcal{X} \times \mathcal{Y}$. Denote the set of random categorical variables as $\mathcal{C}$, and take its subset $\mathscr{E} \subseteq \mathcal{C}$. Then $\mathscr{E}$ is an environment set iff there exists $\mathcal{E} \in \mathscr{E}$ such that $X, Y \perp\!\!\!\perp \mathcal{E}$. $\mathcal{E} \in \mathscr{E}$ is called an environment variable. The interaction heterogeneity between $X$ and $Y$ w.r.t. the environment set $\mathscr{E}$ is defined as:*

$$\mathcal{H}^\mathscr{E}(X, Y) = \sup_{\mathcal{E} \in \mathscr{E}} \mathbb{I}(Y; X|\mathcal{E}) - \mathbb{I}(Y; X). \tag{6}$$

Each environment variable $\mathcal{E}$ represents a stochastic 'partition' of $\mathcal{X} \times \mathcal{Y}$, and the condition for the environment set implies that the joint distribution of $X, Y$ could be preserved in each environment. In information theory, $\mathbb{I}(Y; X|\mathcal{E}) - \mathbb{I}(Y; X)$ is called the *interaction information*, which measures the influence of the environment variable $\mathcal{E}$ on the amount of information shared between the target $Y$ and the covariate $X$. And the *interaction heterogeneity* defined in Equation 6 quantifies the *maximal* additional information that can be gained from involving or uncovering the environment variable $\mathcal{E}$. Intuitively, large $\mathcal{H}^\mathscr{E}(P)$ indicates that the predictive power from $X$ to $Y$ is enhanced by $\mathcal{E}$, which means that uncovering the latent sub-population associated with the environment partition $\mathcal{E}$ will benefit the $X \to Y$ prediction.

### 3.2 PREDICTIVE HETEROGENEITY

Based on the mutual information, the computation of the interaction heterogeneity is quite hard, since the standard mutual information is notoriously difficult to estimate especially in big data scenarios. Also, even if the mutual information could be accurately estimated, the prediction model may not be able to make good use of it.

Inspired by Xu et al. (2020), we raise the *Predictive Heterogeneity*, which measures the interaction heterogeneity that can be captured under computational constraints and affects the prediction of models within the specified predictive family. To begin with, we propose the *Conditional Predictive $\mathcal{V}$-information*, which generalizes the predictive $\mathcal{V}$-information.

**Definition 4** (Conditional Predictive $\mathcal{V}$-information). *Let $X, Y$ be two random variables taking values in $\mathcal{X} \times \mathcal{Y}$ and $\mathcal{E}$ be an environment variable. The conditional predictive $\mathcal{V}$-information is defined as:*

$$\mathbb{I}_\mathcal{V}(X \to Y|\mathcal{E}) = H_\mathcal{V}(Y|\emptyset, \mathcal{E}) - H_\mathcal{V}(Y|X, \mathcal{E}), \tag{7}$$

*where $H_\mathcal{V}(Y|\emptyset, \mathcal{E})$ and $H_\mathcal{V}(Y|X, \mathcal{E})$ are defined as:*

$$H_\mathcal{V}(Y|X, \mathcal{E}) = \mathbb{E}_{e \sim \mathcal{E}} \left[ \inf_{f \in \mathcal{V}} \mathbb{E}_{x, y \sim X, Y|\mathcal{E}=e}[-\log f[x](y)] \right]. \tag{8}$$

$$H_\mathcal{V}(Y|\emptyset, \mathcal{E}) = \mathbb{E}_{e \sim \mathcal{E}} \left[ \inf_{f \in \mathcal{V}} \mathbb{E}_{y \sim Y|\mathcal{E}=e}[-\log f[\emptyset](y)] \right]. \tag{9}$$

Intuitively, the conditional predictive $\mathcal{V}$-information measures the weighted average of predictive $\mathcal{V}$-information among environments. And here we are ready to formalize the predictive heterogeneity measure.

**Definition 5** (Predictive Heterogeneity). *Let $X, Y$ be random variables taking values in $\mathcal{X} \times \mathcal{Y}$ and $\mathscr{E}$ be an environment set. The predictive heterogeneity for the prediction $X \to Y$ with respect to $\mathscr{E}$ is defined as:*

$$\mathcal{H}_\mathcal{V}^\mathscr{E}(X \to Y) = \sup_{\mathcal{E} \in \mathscr{E}} \mathbb{I}_\mathcal{V}(X \to Y|\mathcal{E}) - \mathbb{I}_\mathcal{V}(X \to Y), \tag{10}$$

*where $\mathbb{I}_{\mathcal{V}}(X \to Y)$ is the predictive $\mathcal{V}$-information following from Definition 2.*

Leveraging the predictive $\mathcal{V}$-information, the predictive heterogeneity defined in Equation 10 characterizes the *maximal additional information* that *can be used* by the prediction model when involving the environment variable $\mathcal{E}$. It restricts the prediction models in $\mathcal{V}$ and the explored additional information could benefit the prediction performance of the model $f \in \mathcal{V}$, for which it is named predictive heterogeneity. Next, we present some basic properties of the interaction heterogeneity and predictive heterogeneity.

**Proposition 1** (Basic Properties of Predictive Heterogeneity). *Let $X, Y$ be random variables taking values in $\mathcal{X} \times \mathcal{Y}$, $\mathcal{V}$ be a function family, and $\mathscr{E}$, $\mathscr{E}_1$, $\mathscr{E}_2$ be environment sets.*

1. *Monotonicity: If $\mathscr{E}_1 \subseteq \mathscr{E}_2$, $\mathcal{H}_{\mathcal{V}}^{\mathscr{E}_1}(X \to Y) \leq \mathcal{H}_{\mathcal{V}}^{\mathscr{E}_2}(X \to Y)$.*

2. *Nonnegativity: $\mathcal{H}_{\mathcal{V}}^{\mathscr{E}}(X \to Y) \geq 0$.*

3. *Boundedness: $\mathcal{H}_{\mathcal{V}}^{\mathscr{E}}(X \to Y) \leq H_{\mathcal{V}}(Y|X)$.*

4. *Corner Case: If the predictive family $\mathcal{V}$ is the largest possible predictive family that includes all possible models, i.e. $\mathcal{V} = \Omega$, we have $\mathcal{H}^{\mathscr{E}}(X, Y) = \mathcal{H}_{\Omega}^{\mathscr{E}}(X \to Y)$.*

For further theoretical properties of predictive heterogeneity, in Section 3.3, we derive its explicit forms under *endogeneity*, a common reflection of data heterogeneity. And we demonstrate in Section 3.4 that our proposed predictive heterogeneity can be empirically estimated with guarantees if the complexity of $\mathcal{V}$ is bounded (e.g., its Rademacher complexity).

## 3.3 THEORETICAL PROPERTIES IN LINEAR CASES

In this section, we analyze the theoretical properties of the predictive heterogeneity in multiple linear settings, including (1) a homogeneous case with independent noises and (2) heterogeneous cases with endogeneity brought by selection bias and hidden variables. Under these typical settings, we could approximate the analytical forms of the proposed measure and the conclusions provide insights for general cases.

Firstly, under a homogeneous case with no data heterogeneity, Theorem 1 proves that our measure is bounded by the scale of label noises (which is usually small) and reduces to 0 in linear case under mild assumptions. It indicates that the predictive heterogeneity is insensitive to independent noises. Notably that in the linear case we only deal with the environment variable satisfying $X \perp \epsilon|\mathcal{E}$, since in common prediction tasks, the independent noises are unknown and unrealistic to be exploited for the inference of latent environments $\mathcal{E}$.

**Theorem 1** (Homogeneous Case with Independent Noises). *For a prediction task $X \to Y$ where $X, Y$ are random variables taking values in $\mathbb{R}^n \times \mathbb{R}$, consider the data generation process as $Y = g(x) + \epsilon, \epsilon \sim \mathcal{N}(0, \sigma^2)$ where $g : \mathbb{R}^n \to \mathbb{R}$ is a measurable function. 1) For a function class $\mathcal{G}$ such that $g \in \mathcal{G}$, define the function family as $\mathcal{V}_{\mathcal{G}} = \{f|f[x] = \mathcal{N}(\phi(x), \sigma_V^2), \phi \in \mathcal{G}, \sigma_V \in \mathbb{R}^+\}$. With an environment set $\mathscr{E}$, we have $\mathcal{H}_{\mathcal{V}_{\mathcal{G}}}^{\mathscr{E}}(X \to Y) \leq \pi\sigma^2$. 2) Take $n = 1$ and $g(x) = \beta x, \beta \in \mathbb{R}$. Assume $\mathbb{E}[X] = 0$ and $\mathbb{E}[X^2]$ exists. Given the function family $\mathcal{V}_{\sigma} = \{f|f[x] = \mathcal{N}(\theta x, \sigma^2), \theta \in \mathbb{R}, \sigma \text{ fixed }\}$ and the environment set $\mathscr{E} = \{\mathcal{E}|\mathcal{E} \in \mathcal{C}, |supp(\mathcal{E})| = 2, X \perp \epsilon|\mathcal{E}\}$. We have $\mathcal{H}_{\mathcal{V}_{\sigma}}^{\mathscr{E}}(X \to Y) = 0$.*

Secondly, we examine the proposed measure under *two typical cases of data heterogeneity* (Fan et al., 2014), named *endogeneity by selection bias* (Heckman, 1979; Winship & Mare, 1992; Cui & Athey, 2022) and *endogeneity with hidden variables* (Fan et al., 2014; Arjovsky et al., 2019).

To begin with, in Theorem 2, we consider the prediction task $X \to Y$ with $X, Y$ taking values in $\mathbb{R}^2 \times \mathbb{R}$. Let $X = [S, V]^T$. The predictive family is specified as:

$$\mathcal{V} = \{f|f[x] = \mathcal{N}(\theta_S S + \theta_V V, \sigma^2), \quad \theta_S, \theta_V \in \mathbb{R}, \sigma = 1\}. \tag{11}$$

And the data distribution $P(X, Y)$ is a mixture of latent sub-populations, which could be formulated by an environment variable $\mathcal{E}^* \in \mathcal{C}$ such that $P(X, Y) = \sum_{e \in \text{supp}(\mathcal{E}^*)} P(\mathcal{E}^* = e)P(X, Y|\mathcal{E}^* = e)$. For each $e \in \text{supp}(\mathcal{E}^*)$, $P(X, Y|\mathcal{E}^* = e)$ is the distribution of a homogeneous sub-population. Note that the prediction task is to predict $Y$ with covariates $X$, and the sub-population structure is latent. That is, $P(\mathcal{E}^*|X, Y)$ is *unknown* for models. In the following, we derive the analytical forms of our measure under the one typical case.

**Theorem 2** (Endogeneity with Selection Bias). *For the prediction task $X = [S, V]^T \rightarrow Y$ with a latent environment variable $\mathcal{E}^*$, the data generation process with selection bias is defined as:*

$$Y = \beta S + f(S) + \epsilon_Y, \epsilon_Y \sim \mathcal{N}(0, \sigma_Y^2); \quad V = r(\mathcal{E}^*) f(S) + \sigma(\mathcal{E}^*) \cdot \epsilon_V, \epsilon_V \sim \mathcal{N}(0, 1), \quad (12)$$

*where $f : \mathbb{R} \rightarrow \mathbb{R}$ and $r, \sigma : supp(\mathcal{E}^*) \rightarrow \mathbb{R}$ are measurable functions. $\beta \in \mathbb{R}$. Assume that $\mathbb{E}[f(S)S] = 0$ and there exists $L > 1$ such that $L\sigma^2(\mathcal{E}^*) < r^2(\mathcal{E}^*)\mathbb{E}[f^2]$. For the predictive family defined in equation 11 and the environment set $\mathscr{E} = \mathcal{C}$, the predictive heterogeneity of the prediction task $[S, V]^T \rightarrow Y$ approximates to:*

$$\mathcal{H}_\mathcal{V}^\mathcal{C}(X \rightarrow Y) \approx \frac{Var(r_e)\mathbb{E}[f^2] + \mathbb{E}[\sigma^2(\mathcal{E}^*)]}{\mathbb{E}[r_e^2]\mathbb{E}[f^2] + \mathbb{E}[\sigma^2(\mathcal{E}^*)]} \mathbb{E}[f^2(S)], error\ bounded\ by\ \frac{1}{2}\max(\sigma_Y^2, R(r, \sigma, f)). \quad (13)$$

*And $R(r(\mathcal{E}^*), \sigma(\mathcal{E}^*), f) = \mathbb{E}[(\frac{1}{\frac{r^2\mathbb{E}[f^2]}{\sigma^2} + 1})^2]\mathbb{E}[f^2] + \mathbb{E}_{\mathcal{E}^*}[(\frac{1}{\frac{r}{\sigma} + \frac{\sigma}{r\mathbb{E}[f^2]}})^2] < \mathbb{E}[f^2](\frac{1}{(L+1)^2} + \frac{1}{L+2+\frac{1}{L}})$.*

Intuitively, the data generation process in Theorem 2 introduces the spurious correlation between the spurious feature $V$ and the target $Y$, which varies across different sub-populations (i.e. $r(\mathcal{E}^*)$ and $\sigma(\mathcal{E}^*)$ varies) and brings about data heterogeneity. Here $\mathbb{E}[f(S)S] = 0$ indicates a model misspecification since there is a nonlinear term $f(S)$ that could not be inferred by the linear predictive family with the stable feature $S$. The constant $L$ characterizes the strength of the spurious correlation between $V$ and $Y$. Larger $L$ means $V$ could provide more information for prediction.

From the approximation in Equation 13, we can see that our proposed predictive heterogeneity is dominated by two terms: (1) $Var[r(\mathcal{E}^*)]/\mathbb{E}[r^2(\mathcal{E}^*)]$ characterizes the variance of $r(\mathcal{E}^*)$ among sub-populations; (2) $\mathbb{E}[f^2(S)]$ reflects the strength of model misspecifications. These two components account for two sources of the data heterogeneity under selection bias, which validates the rationality of our proposed measure. According to the theorem, the more various $r(\mathcal{E}^*)$ among the sub-populations and stronger model misspecifications, the larger the predictive heterogeneity.

In general, Theorem 1 and 2 indicate that (1) our proposed measure is insensitive to the homogeneous cases and (2) for the two typical sources of data heterogeneity, our measure accounts for the key components reflecting the latent heterogeneity. Therefore, the theoretical results validate the rationality of our measure.

### 3.4 PAC Guarantees for Predictive Heterogeneity Estimation

Defined under explicit computation constraints, our Predictive Heterogeneity could be empirically estimated with guarantees if the complexity of the model family $\mathcal{V}$ is bounded. In this work, we provide finite sample generalization bounds with the Rademacher complexity. First, we describe the definition of the empirical predictive heterogeneity, the explicit formula for which could be found in Definition 7 in Appendix.

**Definition 6** (Empirical Predictive Heterogeneity (*informal*)). *For the prediction task $X \rightarrow Y$ with $X, Y$ taking values in $\mathcal{X} \times \mathcal{Y}$, a dataset $\mathcal{D}$ is independently and identically drawn from the population such that $\mathcal{D} = \{(x_i, y_i)_{i=1}^N \sim X, Y\}$. Given the predictive family $\mathcal{V}$ and the environment set $\mathscr{E}_K = \{\mathcal{E}|\mathcal{E} \in \mathcal{C}, supp(\mathcal{E}) = K\}$ where $K \in \mathbb{N}^+$ is the number of environments, the **empirical predictive heterogeneity** $\hat{\mathcal{H}}_\mathcal{V}^{\mathscr{E}_K}(X \rightarrow Y; \mathcal{D})$ with respect to $\mathcal{D}$ is readily obtained by estimating $\mathcal{H}_\mathcal{V}^{\mathscr{E}_K}(X \rightarrow Y)$ on $\mathcal{D}$ with expectations replaced by statistics of finite samples. The formal definition is placed in Definition 7.*

**Theorem 3** (PAC Bound). *Consider the prediction task $X \rightarrow Y$ where $X, Y$ are random variables taking values in $\mathcal{X} \times \mathcal{Y}$. Assume that the predictive family $\mathcal{V}$ satisfies $\forall x \in \mathcal{X}, \forall y \in \mathcal{Y}, \forall f \in \mathcal{V}$, $\log f[x](y) \in [-B, B]$ where $B > 0$. For given $K \in \mathbb{N}$, the environment set is defined as $\mathscr{E}_K = \{\mathcal{E}|\mathcal{E} \in \mathcal{C}, supp(\mathcal{E}) = K\}$. Let $\mathcal{Q}$ be the set of all probability distributions of $X, Y, \mathcal{E}$ where $\mathcal{E} \in \mathscr{E}_K$. Take an $e \in supp(\mathcal{E})$ and define a function class $\mathcal{G}_\mathcal{V} = \{g|g(x, y) = \log f[x](y)Q(\mathcal{E} = e|x, y), f \in \mathcal{V}, Q \in \mathcal{Q}\}$. Denote the Rademacher complexity of $\mathcal{G}$ with $N$ samples by $\mathscr{R}_N(\mathcal{G})$. Then for any $\delta \in (0, 1/(2K+2))$, with a probability over $1 - 2(K+1)\delta$, for dataset $\mathcal{D}$ independently and identically drawn from $X, Y$, we have:*

$$|\mathcal{H}_\mathcal{V}^{\mathscr{E}_K}(X \rightarrow Y) - \hat{\mathcal{H}}_\mathcal{V}^{\mathscr{E}_K}(X \rightarrow Y; \mathcal{D})| \leq 4(K+1)\mathscr{R}_{|\mathcal{D}|}(\mathcal{G}_\mathcal{V}) + 2(K+1)B\sqrt{2\log\frac{1}{\delta}/|\mathcal{D}|}, \quad (14)$$

*where $\mathscr{R}_{|\mathcal{D}|}(\mathcal{G}_\mathcal{V}) = \mathcal{O}(|\mathcal{D}|^{-\frac{1}{2}})$ (Bartlett & Mendelson, 2002).*

## 4 ALGORITHM

To empirically estimate the predictive heterogeneity in Definition 6, we derive the Information Maximization (IM) algorithm from the formal definition in Equation 33 to infer the distribution of $\mathcal{E}$ that maximizes the empirical predictive heterogeneity $\hat{\mathcal{H}}_{\mathcal{V}}^{\mathscr{E}_K}(X \to Y; \mathcal{D})$.

**Objective Function.** Given dataset $\mathcal{D} = \{X_N, Y_N\} = \{(x_i, y_i)\}_{i=1}^N$, denote $\text{supp}(\mathcal{E}) = \{e_1, \ldots, e_K\}$, we parameterize the distribution of $\mathcal{E}|(X_N, Y_N)$ with weight matrix $W \in \mathcal{W}_K$, where $K$ is the pre-defined number of environments and $\mathcal{W}_K = \{W : W \in \mathbb{R}_+^{N \times K} \text{ and } W\mathbf{1}_K = \mathbf{1}_N\}$ is the allowed weight space. Each element $w_{ij}$ in $W$ represents $P(\mathcal{E} = e_j | x_i, y_i)$ (the probability of the $i$-th data point belonging to the $j$-th sub-population). For a predictive family $\mathcal{V}$, the solution to the supremum problem in the Definition 7 is equivalent to the following objective function:

$$\min_{W \in \mathcal{W}_K} \mathcal{R}_{\mathcal{V}}(W, \theta_1^*(W), \ldots, \theta_K^*(W)) = \left\{ \frac{1}{N} \sum_{i=1}^N \sum_{j=1}^K w_{ij} \ell_{\mathcal{V}}(f_{\theta_j^*}(x_i), y_i) + U_{\mathcal{V}}(W, Y_N) \right\},$$

$$\text{s.t.} \quad \theta_j^*(W) \in \arg\min_\theta \left\{ \mathcal{L}_{\mathcal{V}}(W, \theta) = \sum_{i=1}^N w_{ij} \ell_{\mathcal{V}}(f_\theta(x_i), y_i) \right\}, \quad \text{for } j = 1, \ldots, K,$$

(15)

where $f_\theta : \mathcal{X} \to \mathcal{Y}$ denotes a predicting function parameterized by $\theta$, $\ell_{\mathcal{V}}(\cdot, \cdot) : \mathcal{Y} \times \mathcal{Y} \to \mathbb{R}$ represents a loss function and $U_{\mathcal{V}}(W, Y_N)$ is a regularizer. Specifically, $f_\theta$, $\ell_{\mathcal{V}}$ and $U_{\mathcal{V}}$ are determined by the predictive family $\mathcal{V}$. Here we provide implementations for two typical and general machine learning tasks, regression and classification.

**(1)** For the *regression task*, the predictive family is typically modeled as:

$$\mathcal{V}_1 = \{g : g[x] = \mathcal{N}(f_\theta(x), \sigma^2), f \text{ is the predicting function and } \theta \text{ is learnable}, \sigma \text{ is a constant}\}. \quad (16)$$

The corresponding loss function is $\ell_{\mathcal{V}_1}(f_\theta(X), Y) = (f_\theta(X) - Y)^2$, and $U_{\mathcal{V}_1}(W, Y_N)$ becomes

$$U_{\mathcal{V}_1}(W, Y_N) = \text{Var}_{j \in [K]}(\overline{Y_N^j}) = \sum_{j=1}^K \left( \sum_{i=1}^N w_{ij} y_i \right)^2 \frac{1}{N \sum_{i=1}^N w_{ij}} - \left( \frac{1}{N} \sum_{i=1}^N y_i \right)^2 \quad (17)$$

where $\overline{Y_N^j}$ denotes the mean value of the label $Y$ given $\mathcal{E} = e_j$ and $U(W, Y_N)$ calculates the variance of $\overline{Y_N^j}$ among sub-populations $e_1 \sim e_K$.

**(2)** For the *classification task*, the predictive family is typically modeled as:

$$\mathcal{V}_2 = \{g : g[x] = f_\theta(x) \in \Delta_c, f \text{ is the classification model and } \theta \text{ is learnable}\}, \quad (18)$$

where $c$ is the class number and $\Delta_c$ denotes the $c$-dimensional simplex. Here each model in the predictive family $\mathcal{V}_2$ outputs a discrete distribution in the form of a $c$-dimensional simplex. In this case, the corresponding loss function $\ell_{\mathcal{V}_2}(\cdot, \cdot)$ is the cross entropy loss and the regularizer becomes $U_{\mathcal{V}_2}(W, Y_N) = -\sum_{j=1}^K \frac{1}{N} (\sum_{i=1}^N w_{ij}) H(Y_N^j)$, where $H(Y_N^j)$ is the entropy of $Y$ given $\mathcal{E} = e_j$.

**Optimization.** The bi-level optimization in Equation 15 can be solved by performing projected gradient descent w.r.t. $W$. The gradient of $W$ can be calculated by: (we omit the subscript $\mathcal{V}$ here)

$$\nabla_W \mathcal{R} = \nabla_W U + \left[ \ell(f_{\theta_j}(x_i), y_i) \right]_{i,j}^{N \times K} + \sum_{j=1}^K \boxed{\nabla_{\theta_j} \mathcal{R}|_{\theta_j^*} \nabla_W \theta_j^*}, \quad (19)$$

$$\text{where} \quad \boxed{\nabla_{\theta_j} \mathcal{R}|_{\theta_j^*} \nabla_W \theta_j^*} \approx \nabla_{\theta_j} \mathcal{R}|_{\theta_j^t} \sum_{h \leq t} \left[ \prod_{k < h} (I - \frac{\partial^2 \mathcal{L}}{\partial \theta_j \partial \theta_j^{\mathrm{T}}}\Big|_{\theta_j^{t-k-1}}) \right] \frac{\partial^2 \mathcal{L}}{\partial \theta_j \partial W^{\mathrm{T}}}\Big|_{\theta_j^{t-h-1}} \quad (20)$$

$$\approx \nabla_{\theta_j} \mathcal{R}|_{\theta_j^t} \frac{\partial^2 \mathcal{L}}{\partial \theta_j \partial W^{\mathrm{T}}}\Big|_{\theta_j^{t-1}} \quad , \text{for } j = 1, \ldots, K. \quad (21)$$

where $\ell(f_{\theta_j}(x_i), y_i)]_{i,j}^{N \times K}$ is an $N \times K$ matrix in which the $(i, j)$-th element is $\ell(f_{\theta_j}(x_i), y_i)$. Here Equation 20 approximates $\theta_j^*$ by $\theta_j^t$ from $t$ steps of inner loop gradient descent and Equation 21 performs *1-step truncated backpropagation* (Shaban et al., 2019; Zhou et al., 2022). Our information maximization algorithm updates $W$ by projected gradient descent as:

$$W \leftarrow \text{Proj}_{\mathcal{W}_K}(W - \eta \nabla_W \mathcal{R}), \quad \eta \text{ is the learning rate of } W. \quad (22)$$

Then we prove that minimizing Equation 15 exactly finds the supremum w.r.t. $\mathcal{E}$ in the Definition 7 (formal) of the empirical predictive heterogeneity.

**Theorem 4** (Justification of the IM Algorithm). *For the regression task with predictive family $\mathcal{V}_1$ and classification task with $\mathcal{V}_2$, the optimization of Equation 15 is equivalent to the supremum problem of the empirical predictive heterogeneity $\hat{\mathcal{H}}_{\mathcal{V}_1}^{\mathscr{E}_K}(X \rightarrow Y; \mathcal{D})$, $\hat{\mathcal{H}}_{\mathcal{V}_2}^{\mathscr{E}_K}(X \rightarrow Y; \mathcal{D})$ respectively in Equation 33 with the pre-defined environment number $K$ (i.e. $supp(\mathcal{E}) = K$).*

**Remark 1** (Difference from Expectation Maximization). *The expectation maximization (EM) algorithm is to infer latent variables of a statistic model to achieve the **maximum likelihood**. Our proposed information maximization (IM) algorithm is to infer the latent variables $W$ which brings the **maximal predictive heterogeneity** associated with the maximal information. Due to the regularizer $U_{\mathcal{V}}$ in our objective function, the EM algorithm cannot efficiently solve our problem, and therefore we adopt bi-level optimization techniques.*

## 5 EXPERIMENTS

### 5.1 PROVIDE INSIGHTS FOR THE SUB-POPULATION DIVISION

The predictive heterogeneity could provide insights for the sub-population division and benefit decision-making, and we illustrate this in prediction tasks of various fields, including agricultural research, sociological research, and object recognition. From the illustrative examples, we show that the learned sub-population division is highly explainable and relevant to decision-making.

**Example: Agriculture**  It is known that the climate affects crop yields and crop suitability (Lobell et al., 2008). We leverage the data from the NOAA database which contains daily weather from weather stations around the world. Following Zhao et al. (2021), we summarize the weather sequence of the year 2018 into summary statistics, including the average yearly temperature, humidity, wind speed and rainy days. The task is to predict the *crop yield* in each place with *weather summary statistics* and *location covariates (i.e. longitude and latitude)* of the place. For easy illustration, we focus on the places with crop types of wheat or rice. Notably that the input covariates do *not* contain the crop type. We use MLP models in this task and set $K = 2$ for our IM algorithm.

Since the crop yield prediction mechanisms are closely related to the crop type that is unknown in the prediction task, we think this causes data heterogeneity in the entire data and the recognized predictive heterogeneity should relate to it. In Figure 1 (a), we plot the real distribution map of wheat and rice planting areas, and in Figure 1 (b), we plot the learned two sub-populations of our IM algorithm. From the results, we surprisingly find the division given by our algorithm is quite similar to the real division of the two crops, indicating the rationality of our measure. For the areas that are not similar (e.g. Tibet Plateau in Asia), we think it is due to some missing features (e.g. population density, altitude) that significantly affect the crop yields.

**Example: Sociology**  We use the UCI Adult dataset (Kohavi & Becker, 1996), which is derived from the 1994 Current Population Survey conducted by the US Census Bureau and is widely used in the study of algorithmic fairness. The task is to predict whether the income of a person is greater or less than 50k US dollars according to personal features. We use linear models in this task and set $K = 2$. In this example, we would like to investigate whether there exist *sub-population structures* inside data that affect the learning of machine learning models.

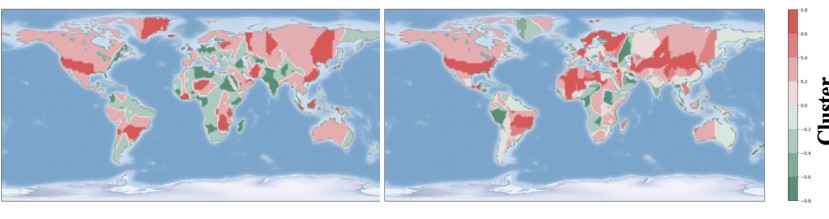

(a) Division of wheat and rice cultivation areas    (b) Division learned by our algorithm

Figure 1: Results on the crop yield data. We color each region according to its main crop type, and the shade represents the proportion of the main crop type after smoothing via $k$-means ($k = 3$).

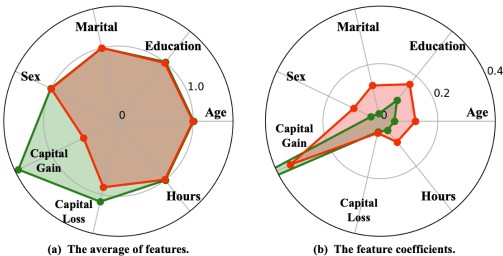

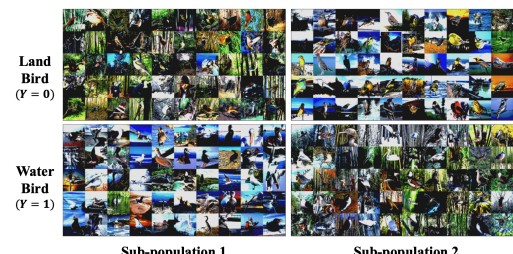

(a) The average of features.  (b) The feature coefficients.  Sub-population 1  Sub-population 2

Figure 2: Results on the Adults data. Here we show the average of features and the feature coefficients of the two learned sub-populations.

Figure 3: Results on the Waterbird data. Here we *randomly sample* 50 images for each class and each learned sub-population.

In Figure 2 (a), we plot summary statistics for the two sub-populations, where the main difference lies in the capital gain. In Figure 2 (b), we plot the feature importance given by linear models for the two sub-populations, and we find that for people with a high capital gain, the prediction model focuses mainly on capital gain, which is fair. However, for people with a low capital gain, models also address some sensitive attributes such as sex and marital status, which tend to cause discrimination. Our results correspond with the results in Zhao et al. (2021) and can help us identify potential inequality in decision-making. For example, our results indicate the potential discrimination for low capital gain people, which could further promote algorithm design and improve policy fairness.

**Example: Object Recognition** Finally, we use the Waterbird dataset (Sagawa et al., 2019), which is widely used to evaluate the model's robustness in the robust learning field. The task is to recognize waterbirds or landbirds. However, the image backgrounds are *spuriously correlated* with the target label, i.e. for the *majority*, waterbirds are on the water and landbirds on the land, and for *minority*, the correlation is reversed. Therefore, the spurious correlation causes predictive heterogeneity in this dataset, since such correlation could affect the machine learning model. In this example, we use the ResNet18 and set $K = 2$ in our IM algorithm. In Figure 3, to show the learned sub-populations of our method, we randomly sample 50 images for each class (waterbird or landbird) and each learned sub-population. In sub-population 1, the majority of landbirds are on the ground and waterbirds are in the water, while in sub-population 2, the majority of landbirds are in the water and waterbirds are on the ground. Our measure captures such spurious correlation and in the two sub-populations, the spurious correlation between the object and background is inverse. And the learned sub-populations could be leveraged by many robust learning methods (Sagawa et al., 2019; Koyama & Yamaguchi, 2020) to learn models with better generalization ability, since they can help to eliminate the influence of backgrounds on model predictions.

## 5.2 BENEFIT OOD GENERALIZATION

The predictive heterogeneity could benefit the out-of-distribution (OOD) generalization of machine learning models. Here we investigate the empirical performance of our IM algorithm w.r.t. the OOD generalization performances on simulated data and real-world colored MNIST data.

**Baselines** First, we compare with *empirical risk minimization* (ERM) and *environment inference for invariant learning* (EIIL, (Creager et al., 2021)) which infers the environments for learning invariance. Then we compare with the well-known *KMeans* algorithm, which is the most popular

Table 1: Results of the experiments on out-of-distribution generalization, including the simulated data and colored MNIST data.

| Method | | 1. Simulated Data | | | | 2. Colored MNIST | |
|---|---|---|---|---|---|---|---|
| | | **Training Sub-population Error** | | **Test Error** | | **Train Accuracy** | **Test Accuracy** |
| | | Major ($r = 1.9$) | Minor ($r = -1.9$) | $r = -2.3$ | $r = -2.7$ | | |
| ERM | | 0.255($\pm$0.024) | 0.740($\pm$0.022) | 0.738($\pm$0.035) | 0.737($\pm$0.023) | 0.998($\pm$0.001) | 0.406($\pm$0.019) |
| EIIL | | **0.164**($\pm$0.014) | 1.428($\pm$0.035) | 1.431($\pm$0.061) | 1.431($\pm$0.046) | 0.812($\pm$0.006) | 0.610($\pm$0.016) |
| KMeans | Balance | 0.231($\pm$0.022) | 0.847($\pm$0.024) | 0.846($\pm$0.039) | 0.845($\pm$0.026) | **0.999**($\pm$0.001) | 0.328($\pm$0.021) |
| | IRM | 0.231($\pm$0.022) | 0.845($\pm$0.024) | 0.844($\pm$0.039) | 0.843($\pm$0.026) | 0.947($\pm$0.004) | 0.259($\pm$0.021) |
| | IGA | 0.235($\pm$0.022) | 0.840($\pm$0.023) | 0.839($\pm$0.038) | 0.838($\pm$0.027) | 0.997($\pm$0.001) | 0.302($\pm$0.021) |
| Ours | Balance | 0.403($\pm$0.041) | **0.423**($\pm$0.016) | **0.416**($\pm$0.022) | **0.416**($\pm$0.014) | 0.749($\pm$0.012) | **0.692**($\pm$0.039) |
| | IRM | 0.391($\pm$0.039) | **0.432**($\pm$0.016) | **0.430**($\pm$0.022) | **0.430**($\pm$0.014) | 0.759($\pm$0.014) | **0.727**($\pm$0.047) |
| | IGA | 0.449($\pm$0.037) | **0.426**($\pm$0.017) | **0.417**($\pm$0.022) | **0.417**($\pm$0.014) | 0.759($\pm$0.012) | **0.713**($\pm$0.034) |

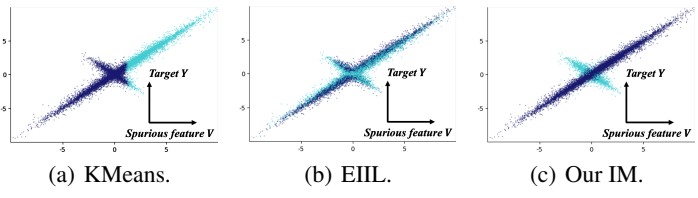

(a) KMeans.    (b) EIIL.    (c) Our IM.

Figure 4: Sub-population division on the simulated data of three methods, where two colors denote two sub-populations.

Figure 5: Sub-population division on the MNIST data of our IM algorithm.

clustering algorithm. For our IM algorithm and KMeans, we involve three algorithms as backbones to leverage the learned sub-populations, including sub-population balancing and invariant learning methods. The sub-population balancing simply equally weighs the learned sub-populations. *invariant risk minimization* (IRM, (Arjovsky et al., 2019)) and *inter-environment gradient alignment* (IGA, (Koyama & Yamaguchi, 2020)) are typical methods in OOD generalization, which take the sub-populations as input environments to learn the invariant models.

**Data Generation of Simulated Data** The input features $X = [S, T, V]^T \in \mathbb{R}^{10}$ consist of stable features $S \in \mathbb{R}^5$, noisy features $T \in \mathbb{R}^4$ and the spurious feature $V \in \mathbb{R}$:

$$S \sim \mathcal{N}(0, 2\mathbf{I}_5), T \sim \mathcal{N}(0, 2\mathbf{I}_4), Y = \theta_S^T S + h(S) + \mathcal{N}(0, 0.5), V \sim \text{Laplace}(\text{sign}(r) \cdot Y, 1/(5 \ln |r|)) \quad (23)$$

where $\theta_S \in \mathbb{R}^5$ is the coefficient and $h(S) = S_1 S_2 S_3$ is the nonlinear term. $|r| > 1$ is a factor for each sub-population, and here the data heterogeneity is brought by the *endogeneity with hidden variable* (Fan et al., 2014). $V$ is the *spurious feature* whose relationship with $Y$ is unstable across sub-populations and is controlled by the factor $r$. Intuitively, $\text{sign}(r)$ controls whether the spurious correlation between $V$ and $Y$ is positive or negative. And $|r|$ controls the strength of the spurious correlation, i.e. the larger $|r|$ means the stronger spurious correlation. In *training*, we generate 10000 points, where the major group contains 80% data with $r = 1.9$ (i.e. strong *positive* spurious correlation) and the minor group contains 20% data with $r = -1.9$ (i.e. strong *negative* spurious correlation). In *testing*, we test the performances of the two groups respectively, and we also set $r = -2.3$ and $r = -2.7$ to simulate stronger distributional shifts. We use linear regression and set $K = 2$ for all methods, and we report the mean-square errors (MSE) of all methods.

**Data Generation of Colored MNIST** Following Arjovsky et al. (2019), we design a binary classification task constructed on the MNIST dataset. Firstly, digits $0 \sim 4$ are labeled $Y = 0$ and digits $5 \sim 9$ are labeled $Y = 1$. Secondly, noisy labels $\tilde{Y}$ are induced by randomly flipping the label $Y$ with a probability of 0.2. Then we sample the colored id $V$ spurious correlated with $\tilde{Y}$ as $V = \begin{cases} +\tilde{Y}, & \text{with probability } r, \\ -\tilde{Y}, & \text{with probability } 1 - r. \end{cases}$. In fact, $r$ controls the spurious correlation between $\tilde{Y}$ and $V$. In *training*, we randomly sample 10000 data points and set $r = 0.85$, meaning that for 85% of the data, $V$ is positively correlated with $\tilde{Y}$ and for the rest 15%, the spurious correlation becomes negative, which causes data heterogeneity w.r.t. $V$ and $\tilde{Y}$. In *testing*, we set $r = 0$ (*strong negative spurious correlation*), bringing strong shifts between training and testing.

**Analysis** From the results in Table 1, for both the simulated and colored MNIST data, the two backbones with our IM algorithm achieve *the best OOD generalization performances*. Also, for the simulated data, the learned predictive heterogeneity enables backbone algorithms to equally treat the majority and minority inside data (i.e. low-performance gap between 'Major' and 'Minor'), and significantly benefits the OOD generalization. Further, for both experiments, we plot the learned sub-populations of our IM algorithm in Figure 4 and 5. From Figure 4, compared with KMeans and EIIL, our predictive heterogeneity exploits the spurious correlation between $V$ and $Y$, and enables the backbone algorithms to eliminate it. From Figure 5, the learned sub-populations of our method also reflect the different directions of the spurious correlation between digit labels $Y$ and colors (red or green), which helps backbone methods to avoid using colors to predict digits.

## 6 CONCLUSION

We define the predictive heterogeneity, as the first quantitative formulation of the data heterogeneity that affects the prediction of machine learning models. We demonstrate its theoretical properties and show that it benefits the out-of-distribution generalization performances.

## ACKNOWLEDGEMENTS

We would like to thank Yuting Pan, Jiaming Song, Fan Bao and anonymous reviewers for helpful feedback. Peng Cui's research was supported in part by National Key R&D Program of China (No. 2018AAA0102004, No. 2020AAA0106300), National Natural Science Foundation of China (No. U1936219, 62141607), Beijing Academy of Artificial Intelligence (BAAI). Bo Li's research was supported by the National Natural Science Foundation of China (No.72171131, 72133002); the Technology and Innovation Major Project of the Ministry of Science and Technology of China under Grants 2020AAA0108400 and 2020AAA0108403.

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

# A FORMAL DEFINITION OF EMPIRICAL PREDICTIVE HETEROGENEITY

In this section, we derive the explicit formula for the empirical estimation of the predictive heterogeneity which is described in Definition 6.

The dataset $\mathcal{D} = \{(x_i, y_i)\}_{i=1}^{|\mathcal{D}|}$ is independently and identically drawn from the population $X, Y$. Given a function family $\mathcal{V}$ and an environment set $\mathscr{E}_K$, let $\mathcal{Q}$ be the set of all probability distributions of $X, Y, \mathcal{E}$ where $\mathcal{E} \in \mathscr{E}_K$. For given $\mathcal{E}$, denote $\text{supp}(\mathcal{E}) = \{(e_k)_{k=1}^K\}$. The empirical predictive heterogeneity $\hat{\mathcal{H}}_{\mathcal{V}}^{\mathscr{E}_K}(X \to Y; \mathcal{D})$ is given by:

$$\hat{\mathcal{H}}_{\mathcal{V}}^{\mathscr{E}_K}(X \to Y; \mathcal{D}) = \sup_{\mathcal{E} \in \mathscr{E}_K} \hat{\mathbb{I}}_{\mathcal{V}}(X \to Y | \mathcal{E}; \mathcal{D}) - \hat{\mathbb{I}}_{\mathcal{V}}(X \to Y; \mathcal{D}) \tag{24}$$

$$= \sup_{\hat{Q} \in \mathcal{Q}} \sum_{k=1}^K \left[ \hat{Q}(\mathcal{E} = e_k) \hat{H}_{\mathcal{V}}(Y | \mathcal{E} = e_k; \mathcal{D}) - \hat{Q}(\mathcal{E} = e_k) \hat{H}_{\mathcal{V}}(Y | X, \mathcal{E} = e_k; \mathcal{D}) \right] \tag{25}$$

$$- [\hat{H}_{\mathcal{V}}(Y; \mathcal{D}) - \hat{H}_{\mathcal{V}}(Y | X; \mathcal{D})]. \tag{26}$$

Specifically,

$$\hat{Q}(\mathcal{E} = e_k) \hat{H}_{\mathcal{V}}(Y | X, \mathcal{E} = e_k; \mathcal{D}) \tag{27}$$

$$= \inf_{f \in \mathcal{V}} \hat{Q}(\mathcal{E} = e_k) \sum_{x_i, y_i \in \mathcal{D}} - \log f[x_i](y_i) \frac{\hat{Q}(x_i, y_i | \mathcal{E} = e_k)}{\sum_{x_j, y_j \in \mathcal{D}} \hat{Q}(x_j, y_j | \mathcal{E} = e_k)} \tag{28}$$

$$= \inf_{f \in \mathcal{V}} \hat{Q}(\mathcal{E} = e_k) \sum_{x_i, y_i \in \mathcal{D}} - \log f[x_i](y_i) \frac{\hat{Q}(\mathcal{E} = e_k | x_i, y_i) \hat{Q}(x_i, y_i)}{\sum_{x_j, y_j \in \mathcal{D}} \hat{Q}(\mathcal{E} = e_k | x_j, y_j) \hat{Q}(x_j, y_j)} \tag{29}$$

$$= \inf_{f \in \mathcal{V}} \hat{Q}(\mathcal{E} = e_k) \sum_{x_i, y_i \in \mathcal{D}} - \log f[x_i](y_i) \frac{\hat{Q}(\mathcal{E} = e_k | x_i, y_i) \hat{Q}(x_i, y_i)}{\hat{Q}(\mathcal{E} = e_k)} \tag{30}$$

$$= \inf_{f \in \mathcal{V}} \sum_{x_i, y_i \in \mathcal{D}} - \log f[x_i](y_i) \hat{Q}(\mathcal{E} = e_k | x_i, y_i) \hat{Q}(x_i, y_i) \tag{31}$$

$$= \inf_{f \in \mathcal{V}} \frac{1}{|\mathcal{D}|} \sum_{x_i, y_i \in \mathcal{D}} - \log f[x_i](y_i) \hat{Q}(\mathcal{E} = e_k | x_i, y_i). \tag{32}$$

The explicit formula for $\hat{Q}(\mathcal{E} = e_k) \hat{H}_{\mathcal{V}}(Y | \mathcal{E} = e_k; \mathcal{D})$, $\hat{H}_{\mathcal{V}}(Y | X; \mathcal{D})$ and $\hat{H}_{\mathcal{V}}(Y; \mathcal{D})$ could be similarly derived. Here we are ready to formally define the empirical predictive heterogeneity.

**Definition 7** (Empirical Predictive Heterogeneity (formal)). *For the prediction task $X \to Y$ with $X$, $Y$ taking values in $\mathcal{X} \times \mathcal{Y}$, a dataset $\mathcal{D}$ is independently and identically drawn from the population such that $\mathcal{D} = \{(x_i, y_i)_{i=1}^N \sim X, Y\}$. Given the predictive family $\mathcal{V}$ and the environment set $\mathscr{E}_K = \{\mathcal{E} | \mathcal{E} \in \mathcal{C}, \text{supp}(\mathcal{E}) = K\}$ where $K \in \mathbb{N}$, let $\mathcal{Q}$ be the set of all probability distributions of $X, Y, \mathcal{E}$ where $\mathcal{E} \in \mathscr{E}_K$. The empirical predictive heterogeneity $\hat{\mathcal{H}}_{\mathcal{V}}^{\mathscr{E}_K}(X \to Y; \mathcal{D})$ with respect to $\mathcal{D}$ is defined as:*

$$\hat{\mathcal{H}}_{\mathcal{V}}^{\mathscr{E}_K}(X \to Y; \mathcal{D}) = \sup_{\hat{Q} \in \mathcal{Q}} \sum_{k=1}^K \left[ \hat{Q}(\mathcal{E} = e_k) \hat{H}_{\mathcal{V}}(Y | \mathcal{E} = e_k; \mathcal{D}) - \hat{Q}(\mathcal{E} = e_k) \hat{H}_{\mathcal{V}}(Y | X, \mathcal{E} = e_k; \mathcal{D}) \right]$$
$$- [\hat{H}_{\mathcal{V}}(Y; \mathcal{D}) - \hat{H}_{\mathcal{V}}(Y | X; \mathcal{D})], \tag{33}$$

*where*

$$\hat{Q}(\mathcal{E} = e_k)\hat{H}_\mathcal{V}(Y|X, \mathcal{E} = e_k; \mathcal{D}) = \inf_{f \in \mathcal{V}} \frac{1}{|\mathcal{D}|} \sum_{x_i, y_i \in \mathcal{D}} -\log f[x_i](y_i)\hat{Q}(\mathcal{E} = e_k|x_i, y_i). \quad (34)$$

$$\hat{Q}(\mathcal{E} = e_k)\hat{H}_\mathcal{V}(Y|\mathcal{E} = e_k; \mathcal{D}) = \inf_{f \in \mathcal{V}} \frac{1}{|\mathcal{D}|} \sum_{x_i, y_i \in \mathcal{D}} -\log f[\emptyset](y_i)\hat{Q}(\mathcal{E} = e_k|x_i, y_i). \quad (35)$$

$$\hat{H}_\mathcal{V}(Y|X; \mathcal{D}) = \inf_{f \in \mathcal{V}} \frac{1}{|\mathcal{D}|} \sum_{x_i, y_i \in \mathcal{D}} -\log f[x_i](y_i). \quad (36)$$

$$\hat{H}_\mathcal{V}(Y; \mathcal{D}) = \inf_{f \in \mathcal{V}} \frac{1}{|\mathcal{D}|} \sum_{x_i, y_i \in \mathcal{D}} -\log f[\emptyset](y_i). \quad (37)$$

## B   SENSITIVITY OF $K$

In the experiments of Section 5, we set the $K = 2$ for easy illustrations. In this section, we add the results of choosing different $K$s for the simulated experiment in Section 5.2 to show that the OOD generalization performances of some typical algorithms plus our proposed method are not sensitive to the choices of $K$.

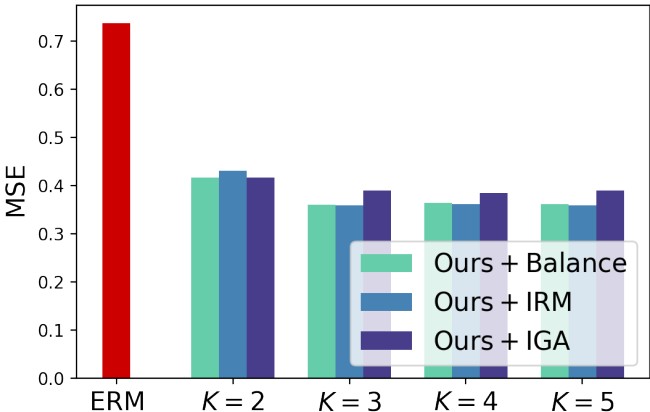

Figure 6: The out-of-distribution generalization error of our methods with Sub-population Balancing, IRM and IGA as backbones. Here we plot the errors of different backbones under $r = -2.7$, which introduces strong distributional shifts with training data.

In Figure 6, we show the out-of-distribution generalization error of our methods with Sub-population Balancing, IRM and IGA as backbones. We plot the OOD testing performances under $r = -2.7$, which has strong distributional shift with the training distribution. From the results, we can see that the performances of three OOD generalization methods *do not be affected much* by the choice of $K$, and from Table 1 , our performances significantly outperforms all the baselines.

Also, we add one more experiment to show that (1) when the chosen $K$ is smaller than the ground-truth, the performances of our methods will drop but are still better than ERM (2) when the chosen $K$ is larger, the performances are not affected much (consistent with the results in Appendix B).

Experiment Setting: The input features $X = [S, T, V] \in \mathbb{R}^{10}$ consist of stable features $S \in \mathbb{R}^5$, noisy features $T \in \mathbb{R}^4$ and the spurious feature $V \in \mathbb{R}$:

$$S \sim \mathcal{N}(2, 2\mathbb{I}_5), \quad T \sim \mathcal{N}(0, 2\mathbb{I}_4), \quad Y = \theta_S^T S + S_1 S_2 S_3 + \mathcal{N}(0, 0.5),$$

and we generate the spurious feature via:

$$V = \theta_V^e Y + \mathcal{N}(0, 0.3),$$

where $\theta_V^e$ varies across sub-populations and is dependent on which sub-population the data point belongs to. In training, we sample 8000 data points from $e_1$ with $\theta_V^1 = 3.0$, 1000 points from $e_2$ with $\theta_V^2 = -1.0$, 1000 points from $e_3$ with $\theta_V^3 = -2.0$ and 1000 points from $e_4$ with $\theta_V^4 = -3.0$. Therefore, the ground-truth number of sub-populations is 4. In testing, we test the performances on $e_4$ with $\theta_V^4 = -3.0$, which has strong distributional shifts from training data. The average MSE over 10 runs are shown in Figure 7.

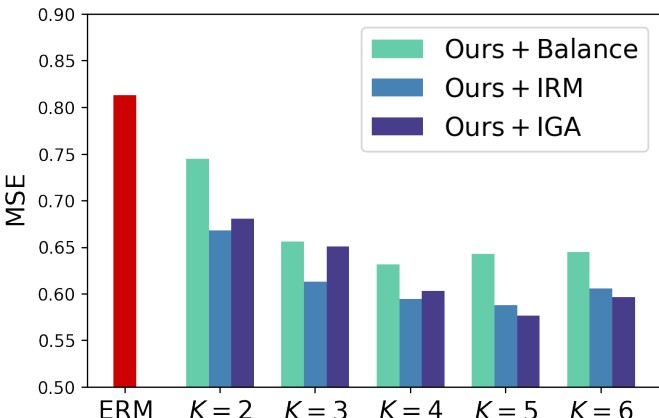

Figure 7: The out-of-distribution generalization error of our methods with Sub-population Balancing, IRM and IGA as backbones for the added experiments. The ground-truth sub-population number is 4.

From the results, we can see that when $K$ is smaller than the ground-truth, increasing $K$ benefits the OOD generalization performance, and when $K$ is larger, the performances are not affected much, which is consistent with the results in Figure 6.

For our IM algorithm, we think there are mainly two ways to choose $K$:

- According to the predictive heterogeneity index: When the chosen $K$ is smaller than the ground-truth, our measure tends to increase quickly when increasing $K$; and when $K$ is larger than the ground-truth, the increasing speed will slow down, which could direct people to choose an appropriate $K$.

- According to the prediction model: Since our IM algorithm aims to learn sub-populations with different prediction mechanisms, one could compare the learned model parameters $\theta_1, \ldots, \theta_K$ to judge whether $K$ is much larger than the ground-truth, i.e., if two resultant models are quite similar, $K$ may be too large (divide one sub-population into two). For linear models, one can directly compare the coefficients. For deep models, we think one can calculate the transfer losses across sub-populations.

For a detailed analysis of the best choice of $K$, we leave it for future work.

## C RELATED WORK

To the best of our knowledge, data heterogeneity has not converged to a uniform formulation so far, and has different meanings among different fields. Li & Reynolds (1995) define the heterogeneity in *ecology* based on the system property and complexity or variability. Rosenbaum (2005) views the uncertainty of the potential outcome as unit heterogeneity in observational studies in *economics*. For *graph* data, the heterogeneity refers to various types of nodes and edges (Wang et al. (2019)). More recently, in machine learning, several works of *causal learning* (Peters et al., 2016; Arjovsky et al., 2019; Koyama & Yamaguchi, 2020; Creager et al., 2021) and *robust learning* (Sagawa et al., 2019) leverage heterogeneous data from multiple environments to improve the out-of-distribution generalization ability. Specifically, invariant learning methods (Arjovsky et al., 2019; Koyama &

Yamaguchi, 2020; Creager et al., 2021; Zhou et al., 2022) leverage the heterogeneous environment to learn the invariant predictors that have uniform performances across environments. And in distributionally robust optimization field, Sagawa et al. (2019); Duchi et al. (2022) propose to optimize the worst-group prediction error to guarantee the OOD generalization performance. However, in machine learning, previous works have not provided a precise definition or sound quantification of data heterogeneity, which makes it confusing and hard to leverage to develop more rational machine learning algorithms.

As for clustering algorithms, most algorithms only focus on the covariates $X$, typified by KMeans and Gaussian Mixture Model (GMM, (Reynolds, 2009)). However, the learned clusters by KMeans cannot reflect the predictive heterogeneity, which is shown by our experiments. And the expectation maximization (EM, (Moon, 1996)) can also be used for clustering. However, our IM algorithm has essential differences from EM, for our IM algorithm infers latent variables that maximizes the predictive heterogeneity but EM maximizes the likelihood. Also, there are methods (Creager et al., 2021) from the invariant learning field to infer environments. Though it could benefit the OOD generalization, it lacks the theoretical foundation and only works in some settings.

## D    PROOF OF PROPOSITION 1

*Proof of Proposition 1.*

1. *Monotonicity*:

Because of $\mathscr{E}_1 \subseteq \mathscr{E}_2$,

$$\mathcal{H}_{\mathcal{V}}^{\mathscr{E}_1}(X \to Y) = \sup_{\mathcal{E} \in \mathscr{E}_1} \mathbb{I}_{\mathcal{V}}(X \to Y | \mathcal{E}) - \mathbb{I}_{\mathcal{V}}(X \to Y) \tag{38}$$

$$\leq \sup_{\mathcal{E} \in \mathscr{E}_2} \mathbb{I}_{\mathcal{V}}(X \to Y | \mathcal{E}) - \mathbb{I}_{\mathcal{V}}(X \to Y) \tag{39}$$

$$= \mathcal{H}_{\mathcal{V}}^{\mathscr{E}_2}(X \to Y). \tag{40}$$

2. *Nonnegativity*:

According to the definition of the environment set, there exists $\mathcal{E}_0 \in \mathscr{E}$ such that for any $e \in \text{supp}(\mathcal{E})$, $X, Y | \mathcal{E} = e$ is identically distributed as $X, Y$. Thus, we have

$$\mathcal{H}_{\mathcal{V}}^{\mathscr{E}}(X \to Y) = \sup_{\mathcal{E} \in \mathscr{E}} \left[ H_{\mathcal{V}}(Y | \emptyset, \mathcal{E}) - H_{\mathcal{V}}(Y | X, \mathcal{E}) \right] - \left[ H_{\mathcal{V}}(Y | \emptyset) - H_{\mathcal{V}}(Y | X) \right] \tag{41}$$

$$\geq \left[ H_{\mathcal{V}}(Y | \emptyset, \mathcal{E}_0) - H_{\mathcal{V}}(Y | X, \mathcal{E}_0) \right] - \left[ H_{\mathcal{V}}(Y | \emptyset) - H_{\mathcal{V}}(Y | X) \right]. \tag{42}$$

Specifically,

$$H_{\mathcal{V}}(Y | X, \mathcal{E}_0) = \mathbb{E}_{e \sim \mathcal{E}_0} \left[ \inf_{f \in \mathcal{V}} \mathbb{E}_{x, y \sim X, Y | \mathcal{E} = e}[- \log f[x](y)] \right] \tag{43}$$

$$= \mathbb{E}_{e \sim \mathcal{E}_0} \left[ \inf_{f \in \mathcal{V}} \mathbb{E}_{x, y \sim X, Y}[- \log f[x](y)] \right] \tag{44}$$

$$= H_{\mathcal{V}}(Y | X). \tag{45}$$

Similarly, $H_{\mathcal{V}}(Y | \emptyset, \mathcal{E}_0) = H_{\mathcal{V}}(Y | \emptyset)$. Thus, $\mathcal{H}_{\mathcal{V}}^{\mathscr{E}}(X \to Y) \geq 0$.

3. *Boundedness*:

First, we have

$$H_{\mathcal{V}}(Y | X, \mathcal{E}) = \mathbb{E}_{e \sim \mathcal{E}} \left[ \inf_{f \in \mathcal{V}} \mathbb{E}_{x, y \sim X, Y | \mathcal{E} = e}[- \log f[x](y)] \right] \tag{46}$$

$$= \mathbb{E}_{e \sim \mathcal{E}} \left[ \inf_{f \in \mathcal{V}} \mathbb{E}_{x \sim X | \mathcal{E} = e} \left[ \mathbb{E}_{y \sim Y | x, e}[- \log f[x](y)] \right] \right] \tag{47}$$

$$\geq 0, \tag{48}$$

by noticing that $\mathbb{E}_{y \sim Y | x}[- \log f[x](y)]$ is the cross entropy between $Y | x, e$ and $f[x]$.

Next,

$$H_{\mathcal{V}}(Y|\emptyset, \mathcal{E}) = \mathbb{E}_{e \sim \mathcal{E}} \left[ \inf_{f \in \mathcal{V}} \mathbb{E}_{y \sim Y|\mathcal{E}=e}[-\log f[\emptyset](y)] \right] \tag{49}$$

$$\leq \inf_{f \in \mathcal{V}} \mathbb{E}_{e \sim \mathcal{E}} \left[ \mathbb{E}_{y \sim Y|\mathcal{E}=e}[-\log f[\emptyset](y)] \right] \tag{50}$$

$$= \inf_{f \in \mathcal{V}} \mathbb{E}_{y \sim Y}[-\log f[\emptyset](y)] \tag{51}$$

$$= H_{\mathcal{V}}(Y|\emptyset), \tag{52}$$

where Equation 50 is due to Jensen's inequality.

Combing the above inequalities,

$$\mathcal{H}_{\mathcal{V}}^{\mathscr{E}}(X \to Y) = \sup_{\mathcal{E} \in \mathscr{E}} [H_{\mathcal{V}}(Y|\emptyset, \mathcal{E}) - H_{\mathcal{V}}(Y|X, \mathcal{E})] - [H_{\mathcal{V}}(Y|\emptyset) - H_{\mathcal{V}}(Y|X)] \tag{53}$$

$$\leq \sup_{\mathcal{E} \in \mathscr{E}} H_{\mathcal{V}}(Y|\emptyset, \mathcal{E}) - [H_{\mathcal{V}}(Y|\emptyset) - H_{\mathcal{V}}(Y|X)] \tag{54}$$

$$\leq H_{\mathcal{V}}(Y|\emptyset) - [H_{\mathcal{V}}(Y|\emptyset) - H_{\mathcal{V}}(Y|X)] \tag{55}$$

$$= H_{\mathcal{V}}(Y|X). \tag{56}$$

4. *Corner Case*:

According to Proposition 2 in Xu et al. (2020),

$$H_{\Omega}(Y|\emptyset) = H(Y). \tag{57}$$

$$H_{\Omega}(Y|X) = H(Y|X). \tag{58}$$

By taking random variables $R, S$ identically distributed as $X, Y|\mathcal{E} = e$ for $e \in \text{supp}(\mathcal{E})$, we have

$$H_{\Omega}(Y|X, \mathcal{E} = e) = H_{\Omega}(S|R) = H(S|R) = H(Y|X, \mathcal{E} = e). \tag{59}$$

Thus,

$$H_{\Omega}(Y|X, \mathcal{E}) = \mathbb{E}_{e \sim \mathcal{E}}[H_{\Omega}(Y|X, \mathcal{E} = e)] = \mathbb{E}_{e \sim \mathcal{E}}[H(Y|X, \mathcal{E} = e)] = H(Y|X, \mathcal{E}). \tag{60}$$

Similarly, we have $H_{\Omega}(Y|\emptyset, \mathcal{E}) = H(Y|\mathcal{E})$. Thus,

$$\mathcal{H}_{\Omega}^{\mathscr{E}}(X \to Y) = \sup_{\mathcal{E} \in \mathscr{E}} [H_{\Omega}(Y|\emptyset, \mathcal{E}) - H_{\Omega}(Y|X, \mathcal{E})] - [H_{\Omega}(Y|\emptyset) - H_{\Omega}(Y|X)] \tag{61}$$

$$= \sup_{\mathcal{E} \in \mathscr{E}} [H(Y|\mathcal{E}) - H(Y|X, \mathcal{E})] - [H(Y) - H(Y|X)] \tag{62}$$

$$= \sup_{\mathcal{E} \in \mathscr{E}} \mathbb{I}(Y; X|\mathcal{E}) - \mathbb{I}(Y; X) \tag{63}$$

$$= \mathcal{H}^{\mathscr{E}}(X, Y). \tag{64}$$

$\square$

# E  PROOF OF THEOREM 1

*Proof of Theorem 1.*

1)

$$H_{\mathcal{V}_{\mathcal{G}}}(Y|X) = \inf_{f \in \mathcal{V}_{\mathcal{G}}} \mathbb{E}_{x \sim X} \left[ \mathbb{E}_{y \sim Y|x}[-\log f[x](y)] \right] \tag{65}$$

$$\leq \mathbb{E}_{x \sim X} \left[ \mathbb{E}_{y \sim Y|x}[-\log \frac{1}{\sqrt{2\pi} \cdot \frac{1}{\sqrt{2\pi}}} \exp\left[-\frac{(y - g(x))^2}{2 \cdot \frac{1}{2\pi}}\right]] \right] \tag{66}$$

$$= \mathbb{E}_{x \sim X} \left[ \mathbb{E}_{y \sim Y|x}[\pi(y - g(x))^2] \right] \tag{67}$$

$$= \pi\sigma^2. \tag{68}$$

Equation 66 holds by taking $f[x] = \mathcal{N}(g(x), \frac{1}{2\pi})$.

2)

Given the function family $\mathcal{V}_\sigma = \{f | f[x] = \mathcal{N}(\theta x, \sigma^2), \theta \in \mathbb{R}, \sigma \text{ fixed }\}$, by expanding the Gaussian probability density function in the definition of predictive $\mathcal{V}$-information, it could be shown that

$$\mathbb{I}_{\mathcal{V}_\sigma}(X \to Y) \propto \min_{k \in \mathbb{R}} \mathbb{E}[(Y - kX)^2] - \text{Var}(Y), \tag{69}$$

where the predictive $\mathcal{V}$-information is proportional to Mean Square Error subtracted by the variance of target, by a coefficient completely dependent on $\sigma$.

The minimization problem is solved by

$$k = \frac{\mathbb{E}[XY]}{\mathbb{E}[X^2]} = 1. \tag{70}$$

Substituting $k$ into eq.69,

$$\mathbb{I}_{\mathcal{V}_\sigma}(X \to Y) \propto \mathbb{E}[\epsilon^2] - \text{Var}(X + \epsilon) \tag{71}$$

$$= -\text{Var}(X) = -\mathbb{E}[X^2]. \tag{72}$$

Denote $\text{supp}(\mathcal{E}) = \{\mathcal{E}_1, \mathcal{E}_2\}$. Let $Q$ be the joint distribution of $(X, \epsilon, \mathcal{E})$. Let $Q(\mathcal{E}_1) = \alpha$ and $Q(\mathcal{E}_2) = 1 - \alpha$ be the marginal of $\mathcal{E}$. Abbreviate $Q(X, \epsilon | \mathcal{E} = \mathcal{E}_1)$ by $P_1(X, \epsilon)$ and $Q(X, \epsilon | \mathcal{E} = \mathcal{E}_2)$ by $P_2(X, \epsilon)$.

Similar to 69,

$$\mathbb{I}_{\mathcal{V}_\sigma}(X \to Y | \mathcal{E}) \propto \min_k \mathbb{E}[(Y - kX)^2 | \mathcal{E}] - \text{Var}(Y | \mathcal{E}). \tag{73}$$

For $\mathcal{E} = \mathcal{E}_1$, the minimization problem is solved by

$$k = \frac{\mathbb{E}_{P_1}[XY]}{\mathbb{E}_{P_1}[X^2]}. \tag{74}$$

Thus,

$$\mathbb{I}_{\mathcal{V}_\sigma}(X \to Y | \mathcal{E} = \mathcal{E}_1) \propto \mathbb{E}_{P_1}\left[\left(Y - \frac{\mathbb{E}_{P_1}[XY]}{\mathbb{E}_{P_1}[X^2]} X\right)^2\right] - \text{Var}_{P_1}(Y) \tag{75}$$

$$= \mathbb{E}_{P_1}[Y^2] - \frac{\mathbb{E}_{P_1}^2[XY]}{\mathbb{E}_{P_1}[X^2]} - (\mathbb{E}_{P_1}[Y^2] - \mathbb{E}_{P_1}^2[Y]) \tag{76}$$

$$= \mathbb{E}_{P_1}^2[Y] - \frac{\mathbb{E}_{P_1}^2[XY]}{\mathbb{E}_{P_1}[X^2]}. \tag{77}$$

Similarly, we have

$$\mathbb{I}_{\mathcal{V}_\sigma}(X \to Y | \mathcal{E} = \mathcal{E}_2) \propto \mathbb{E}_{P_2}^2[Y] - \frac{\mathbb{E}_{P_2}^2[XY]}{\mathbb{E}_{P_2}[X^2]}. \tag{78}$$

Notably, $\mathbb{E}_{P_1}[X^2]$ and $\mathbb{E}_{P_2}[X^2]$ are constrained by $\alpha$ and $\mathbb{E}[X^2]$.

$$\mathbb{E}[X^2] = \mathbb{E}[\mathbb{E}[X^2 | \mathcal{E}]] = \alpha \mathbb{E}_{P_1}[X^2] + (1 - \alpha) \mathbb{E}_{P_2}[X^2]. \tag{79}$$

Similarly,

$$\mathbb{E}[X^2] = \mathbb{E}[XY] = \alpha \mathbb{E}_{P_1}[XY] + (1 - \alpha) \mathbb{E}_{P_2}[XY]. \tag{80}$$

$$0 = \mathbb{E}[Y] = \alpha \mathbb{E}_{P_1}[Y] + (1 - \alpha) \mathbb{E}_{P_2}[Y]. \tag{81}$$

The moments of $P_2$ could thereafter be represented by those of $P_1$.

$$\mathbb{E}_{P_2}[X^2] = \frac{\mathbb{E}[X^2] - \alpha \mathbb{E}_{P_1}[X^2]}{1 - \alpha}. \tag{82}$$

$$\mathbb{E}_{P_2}[XY] = \frac{\mathbb{E}[X^2] - \alpha \mathbb{E}_{P_1}[XY]}{1 - \alpha}. \tag{83}$$

$$\mathbb{E}_{P_2}[Y] = -\frac{\alpha \mathbb{E}_{P_1}[Y]}{1-\alpha}. \tag{84}$$

Substituting to eq.78,

$$\mathbb{I}_{\mathcal{V}_\sigma}(X \to Y | \mathcal{E} = \mathcal{E}_2) \propto \frac{\alpha^2}{(1-\alpha)^2} E_{P_1}^2[Y] - \frac{1}{1-\alpha} \frac{\left(\mathbb{E}[X^2] - \alpha \mathbb{E}_{P_1}[XY]\right)^2}{\mathbb{E}[X^2] - \alpha \mathbb{E}_{P_1}[X^2]}. \tag{85}$$

Thus,

$$\mathcal{H}_{\mathcal{V}_\sigma}^{\mathscr{E}}(X \to Y) = \sup_{\mathcal{E} \in \mathscr{E}} \mathbb{I}_{\mathcal{V}_\sigma}(X \to Y) - \alpha \mathbb{I}_{\mathcal{V}_\sigma}(X \to Y | \mathcal{E} = \mathcal{E}_1) - (1-\alpha) \mathbb{I}_{\mathcal{V}_\sigma}(X \to Y | \mathcal{E} = \mathcal{E}_2) \tag{86}$$

$$\propto \sup_{\mathcal{E} \in \mathscr{E}} -\mathbb{E}[X^2] - \alpha \mathbb{E}_{P_1}^2[Y] + \alpha \frac{\mathbb{E}_{P_1}^2[XY]}{\mathbb{E}_{P_1}[X^2]} - \frac{\alpha^2}{1-\alpha} \mathbb{E}_{P_1}^2[Y] + \frac{\left(\mathbb{E}[X^2] - \alpha \mathbb{E}_{P_1}[XY]\right)^2}{\mathbb{E}[X^2] - \alpha \mathbb{E}_{P_1}[X^2]} \tag{87}$$

$$= \sup_{\mathcal{E} \in \mathscr{E}} -\frac{\alpha}{1-\alpha} \mathbb{E}_{P_1}^2[Y] + \alpha \frac{\left(\mathbb{E}_{P_1}[X^2] - \mathbb{E}_{P_1}[XY]\right)^2}{\mathbb{E}_{P_1}[X^2] \left(\mathbb{E}[X^2] - \alpha \mathbb{E}_{P_1}[X^2]\right)} \mathbb{E}[X^2] \tag{88}$$

$$= \sup_{\mathcal{E} \in \mathscr{E}} -\frac{\alpha}{1-\alpha} \mathbb{E}_{P_1}^2[X+\epsilon] + \alpha \frac{\mathbb{E}_{P_1}^2[X\epsilon]}{\mathbb{E}_{P_1}[X^2] \left(\mathbb{E}[X^2] - \alpha \mathbb{E}_{P_1}[X^2]\right)} \mathbb{E}[X^2]. \tag{89}$$

Assuming $X \perp \epsilon \mid \mathcal{E}$,

$$\mathcal{H}_{\mathcal{V}_\sigma}^{\mathscr{E}}(X \to Y) = \sup_{\mathcal{E} \in \mathscr{E}} -\frac{\alpha}{1-\alpha} \mathbb{E}_{P_1}^2[X+\epsilon] \le 0. \tag{90}$$

From Proposition 1, we have $\mathcal{H}_{\mathcal{V}_\sigma}^{\mathscr{E}}(X \to Y) \ge 0$. Thus, $\mathcal{H}_{\mathcal{V}_\sigma}^{\mathscr{E}}(X \to Y) = 0$. $\qquad \square$

# F  PROOF OF LINEAR CASES (THEOREM 2 AND ??)

*Proof of Theorem 2.*

For the ease of notion, we denote the $r(\mathcal{E}^*)$ as $r_e$, $\sigma(\mathcal{E}^*)$ as $\sigma_e$, and $\sigma(\mathcal{E}^*) \cdot \epsilon_v$ as $\epsilon_e$. And we omit the superscript $\mathcal{C}$ of $\mathcal{H}_{\mathcal{V}}^{\mathcal{C}}$. Firstly, we calculate the $H_{\mathcal{V}}[Y|\emptyset]$ as:

$$H_{\mathcal{V}}[Y|\emptyset] = \frac{1}{2\sigma^2} \mathrm{Var}(Y) + \log \sigma + \frac{1}{2} \log 2\pi, \tag{91}$$

$$H_{\mathcal{V}}[Y|\emptyset, \mathcal{E}^*] = \frac{1}{2\sigma^2} \mathbb{E}_{\mathcal{E}^*}[\mathrm{Var}(Y|\mathcal{E}^*)] + \log \sigma + \frac{1}{2} \log 2\pi. \tag{92}$$

Therefore, we have

$$H_{\mathcal{V}}[Y|\emptyset, \mathcal{E}^*] - H_{\mathcal{V}}[Y|\emptyset] = -\frac{1}{2\sigma^2} \mathrm{Var}(\mathbb{E}[Y|\mathcal{E}^*]) \le 0. \tag{93}$$

As for $H_{\mathcal{V}}[Y|X]$, we have

$$H_{\mathcal{V}}[Y|X] = \inf_{h_S, h_V} \mathbb{E}_{X,Y} \left[ \|Y - (h_S S + h_V V)\|^2 \right] \frac{1}{2\sigma^2} \tag{94}$$

$$= \inf_{h_S, h_V} \mathbb{E}_{X,Y} \left[ \|f(S) + \epsilon_Y - (h_S S + h_V V)\|^2 \right] \frac{1}{2\sigma^2} \tag{95}$$

$$= \inf_{h_S, h_V} \mathbb{E}_{\mathcal{E}^*} \left[ \mathbb{E}[\|f(S) + \epsilon_Y - (h_S S + h_V (r_e f(S) + \epsilon_e))\|^2 | \mathcal{E}^*] \right] \frac{1}{2\sigma^2}, \tag{96}$$

where we let $h_S = h_S - \beta$ here. Then we have

$$2\sigma^2 H_{\mathcal{V}}[Y|X] = \inf_{h_S, h_V} \mathbb{E}_{\mathcal{E}^*} \left[ \mathbb{E}[\|(1 - h_V r_e) f(S) + \epsilon_Y - h_S S - h_V \epsilon_e\|^2 | \mathcal{E}^*] \right] \tag{97}$$

$$= \inf_{h_S, h_V} \mathbb{E}_{\mathcal{E}^*} \left[ \mathbb{E}[\|(1 - h_V r_e) f(S) - h_S S\|^2 | \mathcal{E}^*] \right] + \sigma_Y^2 + h_V^2 \mathbb{E}_{\mathcal{E}^*}[\sigma_e^2], \tag{98}$$

notably that here for $e_i, e_j \in \text{supp}(\mathcal{E}^*)$, we assume $P^{e_i}(S, Y) = P^{e_j}(S, Y)$ (we choose such $\mathcal{E}^*$ as one possible split). And the solution of $h_S, h_V$ is

$$h_S = \frac{\text{Var}(r_e)\mathbb{E}[f^2(S)]\mathbb{E}[f(S)S] + \mathbb{E}[\sigma_e^2]\mathbb{E}[f(S)S]}{\mathbb{E}[r_e^2]\mathbb{E}[f^2(S)]\mathbb{E}[S^2] + \mathbb{E}[\sigma_e^2]\mathbb{E}[S^2] - \mathbb{E}^2[r_e]\mathbb{E}^2[f(S)S]}, \tag{99}$$

$$h_V = \frac{\mathbb{E}[r_e](\mathbb{E}[f^2(S)]\mathbb{E}[S^2] - \mathbb{E}^2[f(S)S])}{\mathbb{E}[r_e^2]\mathbb{E}[f^2(S)]\mathbb{E}[S^2] + \mathbb{E}[\sigma_e^2]\mathbb{E}[S^2] - \mathbb{E}^2[r_e]\mathbb{E}^2[f(S)S]}. \tag{100}$$

According to the assumption that $\mathbb{E}[f(S)S] = 0$, we have

$$h_S = 0, \tag{101}$$

$$h_V = \frac{\mathbb{E}[r(\mathcal{E}^*)]\mathbb{E}[f^2]}{\mathbb{E}[r^2(\mathcal{E}^*)]\mathbb{E}[f^2] + \mathbb{E}[\sigma^2(\mathcal{E}^*)]}. \tag{102}$$

Therefore, we have

$$2\sigma^2 H_\mathcal{V}[Y|X] = \mathbb{E}_{\mathcal{E}^*}[\mathbb{E}[\|(1 - h_V r_e)f(S)\|^2|\mathcal{E}^*]] + \sigma_Y^2 + h_V^2\mathbb{E}_{\mathcal{E}^*}[\sigma_e^2] \tag{103}$$

$$= \frac{\text{Var}(r_e)\mathbb{E}[f^2] + \mathbb{E}[\sigma^2(\mathcal{E}^*)]}{\mathbb{E}[r_e^2]\mathbb{E}[f^2] + \mathbb{E}[\sigma^2(\mathcal{E}^*)]}\mathbb{E}[f^2(S)] + \sigma_Y^2, \tag{104}$$

$$2\sigma^2 H_\mathcal{V}[Y|X, \mathcal{E}^*] = \sigma_Y^2 + \mathbb{E}[(\frac{1}{\frac{r_e^2\mathbb{E}[f^2]}{\sigma_e^2} + 1})^2]\mathbb{E}[f^2] + \mathbb{E}_{\mathcal{E}^*}[(\frac{1}{\frac{r_e}{\sigma_e} + \frac{\sigma_e}{r_e\mathbb{E}[f^2]}})^2]. \tag{105}$$

Note that here we simply set $\sigma = 1$ in the main body. And we have:

$$\mathcal{H}_\mathcal{V}(X \to Y) \approx \frac{\text{Var}(r_e)\mathbb{E}[f^2] + \mathbb{E}[\sigma^2(\mathcal{E}^*)]}{\mathbb{E}[r_e^2]\mathbb{E}[f^2] + \mathbb{E}[\sigma^2(\mathcal{E}^*)]}\mathbb{E}[f^2(S)] \tag{106}$$

The approximation error is bounded by $\frac{1}{2}\max(\sigma_Y^2, R(r(\mathcal{E}^*), \sigma(\mathcal{E}^*), \mathbb{E}[f^2]))$, and $R(r(\mathcal{E}^*), \sigma(\mathcal{E}^*), \mathbb{E}[f^2])$ is defined as:

$$R(r(\mathcal{E}^*), \sigma(\mathcal{E}^*), \mathbb{E}[f^2]) = \mathbb{E}[(\frac{1}{\frac{r^2\mathbb{E}[f^2]}{\sigma_e^2} + 1})^2]\mathbb{E}[f^2] + \mathbb{E}_{\mathcal{E}^*}[(\frac{1}{\frac{r_e}{\sigma_e} + \frac{\sigma_e}{r_e\mathbb{E}[f^2]}})^2] \tag{107}$$

$$\square$$

*Proof of Theorem* **??**. As proved above, we have

$$h_S = \beta + \frac{\mathbb{E}[f(S)S]\left(\text{Var}(r_e)(\mathbb{E}[f^2(S)] + \sigma_Y^2) + \mathbb{E}[\sigma_e^2]\right)}{\mathbb{E}[r_e^2]\mathbb{E}[f^2(S)]\mathbb{E}[S^2] + \mathbb{E}[r_e^2]\sigma_Y^2\mathbb{E}[S^2] + \mathbb{E}[\sigma_e^2]\mathbb{E}[S^2] - \mathbb{E}^2[r_e]\mathbb{E}^2[f(S)S]}, \tag{108}$$

$$h_V = \frac{\mathbb{E}[r_e](\sigma_Y^2 + \mathbb{E}[f^2(S)])\mathbb{E}[S^2] - \mathbb{E}[r_e]\mathbb{E}^2[f(S)S]}{\mathbb{E}[r_e^2]\mathbb{E}[f^2(S)]\mathbb{E}[S^2] + \mathbb{E}[r_e^2]\sigma_Y^2\mathbb{E}[S^2] + \mathbb{E}[\sigma_e^2]\mathbb{E}[S^2] - \mathbb{E}^2[r_e]\mathbb{E}^2[f(S)S]}. \tag{109}$$

- For the model misspecification case, we further assume that (1) $\mathbb{E}[f(S)S] = 0$ and (2) $\mathbb{E}[\sigma_e^2] \ll \mathbb{E}[f^2(S)]\mathbb{E}[S^2]$, and then we have

$$h_S = \beta, \tag{110}$$

$$h_V = \frac{\mathbb{E}[r_e]}{\mathbb{E}[r_e^2]}, \tag{111}$$

and for the heterogeneity, we have

$$\frac{\text{Var}(r_e)}{\mathbb{E}[r_e^2]}(\mathbb{E}[f^2(S)] + \mathbb{E}[\sigma_Y^2]) + h_V^2\mathbb{E}_\mathcal{E}[\sigma_e^2] + \sigma_Y^2 \geq 2\sigma^2\mathcal{H}_\mathcal{V}(X \to Y)$$

$$\geq \frac{\text{Var}(r_e)}{\mathbb{E}[r_e^2]}(\mathbb{E}[f^2(S)] + \mathbb{E}[\sigma_Y^2]) + h_V^2\mathbb{E}_\mathcal{E}[\sigma_e^2] - \mathbb{E}_\mathcal{E}[\frac{1}{r_e^2}\sigma_e^2]. \tag{112}$$

- Without the model misspecification, we assume that $f \equiv 0$, and then we have

$$h_S = \beta, \tag{113}$$

$$h_V = \frac{\mathbb{E}[r_e]\sigma_Y^2}{\mathbb{E}[r_e^2]\sigma_Y^2 + \mathbb{E}[\sigma_e^2]}, \tag{114}$$

and for the heterogeneity we have

$$2\sigma^2 \mathcal{H}_{\mathcal{V}}(X \to Y) \geq \sigma_Y^2(1 - 2h_V \mathbb{E}[r_e] + h_V^2 \mathbb{E}[r_e^2]) + h_V^2 \mathbb{E}[\sigma_e^2] - \mathbb{E}[\frac{1}{r_e^2}\sigma_e^2], \quad (115)$$

$$2\sigma^2 \mathcal{H}_{\mathcal{V}}(X \to Y) \leq \sigma_Y^2(1 - 2h_V \mathbb{E}[r_e] + h_V^2 \mathbb{E}[r_e^2]) + h_V^2 \mathbb{E}[\sigma_e^2]. \quad (116)$$

$\square$

## G  PROOF OF THE ERROR BOUND FOR FINITE SAMPLE ESTIMATION (THEOREM 3)

In this section, we will prove the error bound of estimating the predictive heterogeneity with the empirical predictive heterogeneity. Before the proof of Theorem 3 which is inspired by Xu et al. (2020), we will introduce three lemmas.

**Lemma 1.** *Assume* $\forall x \in \mathcal{X}, \forall y \in \mathcal{Y}, \forall f \in \mathcal{V}$, $\log f[x](y) \in [-B, B]$ *where* $B > 0$. *Define a function class* $\mathcal{G}_{\mathcal{V}}^k = \{g | g(x,y) = \log f[x](y)q(\mathcal{E} = e_k | x, y), f \in \mathcal{V}, q \in \mathcal{Q}\}$. *Denote the Rademacher complexity of* $\mathcal{G}$ *with* $N$ *samples by* $\mathscr{R}_N(\mathcal{G})$. *Define* $\hat{f}_k = \arg \inf_f \frac{1}{|\mathcal{D}|} \sum_{x_i, y_i \in \mathcal{D}} -\log f[x_i](y_i)q(\mathcal{E} = e_k | x_i, y_i)$.

*Then for any* $q \in \mathcal{Q}$, *any* $\delta \in (0, 1)$, *with a probability over* $1 - \delta$, *we have*

$$\left| q(\mathcal{E} = e_k)H_{\mathcal{V}}(Y | X, \mathcal{E} = e_k) - \frac{1}{|\mathcal{D}|} \sum_{x_i, y_i \in \mathcal{D}} -\log \hat{f}_k[x_i](y_i)q(\mathcal{E} = e_k | x_i, y_i) \right| \quad (117)$$

$$\leq 2\mathscr{R}_{|\mathcal{D}|}(\mathcal{G}_{\mathcal{V}}^k) + B\sqrt{\frac{2\log\frac{1}{\delta}}{|\mathcal{D}|}}. \quad (118)$$

*Proof.* Apply McDiarmid's inequality to the function $\Phi(\mathcal{D})$ which is defined as:

$$\Phi(\mathcal{D}) = \sup_{f \in \mathcal{V}, q \in \mathcal{Q}} \left| q(\mathcal{E} = e_k)\mathbb{E}_q\left[ -\log f[x](y) | \mathcal{E} = e_k \right] - \frac{1}{|\mathcal{D}|} \sum_{x_i, y_i \in \mathcal{D}} -\log f[x_i](y_i)q(\mathcal{E} = e_k | x_i, y_i) \right|. \tag{119}$$

Let $\mathcal{D}$ and $\mathcal{D}'$ be two identical datasets except for one data point $x_j \neq x_j'$. We have:

$$\Phi(\mathcal{D}) - \Phi(\mathcal{D}') \tag{120}$$

$$\leq \sup_{f \in \mathcal{V}, q \in \mathcal{Q}} \left[ \left| q(\mathcal{E} = e_k)\mathbb{E}_q\left[ -\log f[x](y) | \mathcal{E} = e_k \right] - \frac{1}{|\mathcal{D}|} \sum_{x_i, y_i \in \mathcal{D}} -\log f[x_i](y_i)q(\mathcal{E} = e_k | x_i, y_i) \right| \tag{121}$$

$$- \left| q(\mathcal{E} = e_k)\mathbb{E}_q\left[ -\log f[x](y) | \mathcal{E} = e_k \right] - \frac{1}{|\mathcal{D}'|} \sum_{x_i', y_i' \in \mathcal{D}'} -\log f[x_i'](y_i')q(\mathcal{E} = e_k | x_i', y_i') \right| \right] \tag{122}$$

$$\leq \sup_{f \in \mathcal{V}, q \in \mathcal{Q}} \left| \frac{1}{|\mathcal{D}|} \sum_{x_i, y_i \in \mathcal{D}} -\log f[x_i](y_i)q(\mathcal{E} = e_k | x_i, y_i) - \frac{1}{|\mathcal{D}'|} \sum_{x_i', y_i' \in \mathcal{D}'} -\log f[x_i'](y_i')q(\mathcal{E} = e_k | x_i', y_i') \right| \tag{123}$$

$$= \sup_{f \in \mathcal{V}, q \in \mathcal{Q}} \frac{1}{|\mathcal{D}|} \left| \log f[x_j](y_j)q(\mathcal{E} = e_k | x_j, y_j) - \log f[x_j'](y_j')q(\mathcal{E} = e_k | x_j', y_j') \right| \tag{124}$$

$$\leq \frac{2B}{|\mathcal{D}|}. \tag{125}$$

According to McDiarmid's inequality, for any $\delta \in (0, 1)$, with a probability over $1 - \delta$, we have:

$$\Phi(\mathcal{D}) \leq \mathbb{E}_{\mathcal{D}}[\Phi(\mathcal{D})] + B\sqrt{\frac{2\log\frac{1}{\delta}}{|\mathcal{D}|}}. \tag{126}$$

Next we derive a bound for $\mathbb{E}_{\mathcal{D}}[\Phi(\mathcal{D})]$.

Consider a dataset $\mathcal{D}'$ independently and identically drawn from $q(X, Y) = P(X, Y)$ with the same size as $\mathcal{D}$. We notice that

$$q(\mathcal{E} = e_k)\mathbb{E}_q\left[-\log f[x](y)|\mathcal{E} = e_k\right] \tag{127}$$
$$= q(\mathcal{E} = e_k)\mathbb{E}_q\left[-\log f[x](y)q(\mathcal{E} = e_k|x, y)|\mathcal{E} = e_k\right] \tag{128}$$
$$= \mathbb{E}_q\left[\mathbb{E}_q\left[-\log f[x](y)q(\mathcal{E} = e_k|x, y)|\mathcal{E} = e_k\right]\right] \tag{129}$$
$$= \mathbb{E}_q\left[-\log f[x](y)q(\mathcal{E} = e_k|x, y)\right] \tag{130}$$
$$= \mathbb{E}_{\mathcal{D}'}\left[-\frac{1}{|\mathcal{D}'|}\sum_{x_i', y_i' \in \mathcal{D}'} -\log f[x_i'](y_i')q(\mathcal{E} = e_k|x_i', y_i')\right]. \tag{131}$$

Thus, $\mathbb{E}_{\mathcal{D}}[\Phi(\mathcal{D})]$ could be reformulated as:

$$\mathbb{E}_{\mathcal{D}}[\Phi(\mathcal{D})] \tag{132}$$

$$= \mathbb{E}_{\mathcal{D}}\left[\sup_{f \in \mathcal{V}, q \in \mathcal{Q}}\left|\mathbb{E}_{\mathcal{D}'}\left[-\frac{1}{|\mathcal{D}'|}\sum_{x_i', y_i' \in \mathcal{D}'} -\log f[x_i'](y_i')q(\mathcal{E} = e_k|x_i', y_i')\right]\right.\right. \tag{133}$$

$$\left.\left.-\frac{1}{|\mathcal{D}|}\sum_{x_i, y_i \in \mathcal{D}} -\log f[x_i](y_i)q(\mathcal{E} = e_k|x_i, y_i)\right|\right] \tag{134}$$

$$\leq \mathbb{E}_{\mathcal{D}}\left[\sup_{f \in \mathcal{V}, q \in \mathcal{Q}}\mathbb{E}_{\mathcal{D}'}\left|-\frac{1}{|\mathcal{D}'|}\sum_{x_i', y_i' \in \mathcal{D}'} -\log f[x_i'](y_i')q(\mathcal{E} = e_k|x_i', y_i')\right.\right. \tag{135}$$

$$\left.\left.-\frac{1}{|\mathcal{D}|}\sum_{x_i, y_i \in \mathcal{D}} -\log f[x_i](y_i)q(\mathcal{E} = e_k|x_i, y_i)\right|\right] \tag{136}$$

$$\leq \mathbb{E}_{\mathcal{D}, \mathcal{D}'}\left[\sup_{f \in \mathcal{V}, q \in \mathcal{Q}}\frac{1}{|\mathcal{D}|}\left|\sum_{x_i, y_i \in \mathcal{D}} \log f[x_i](y_i)q(\mathcal{E} = e_k|x_i, y_i)\right.\right. \tag{137}$$

$$\left.\left.-\sum_{x_i', y_i' \in \mathcal{D}'} \log f[x_i'](y_i')q(\mathcal{E} = e_k|x_i', y_i')\right|\right] \tag{138}$$

$$= \mathbb{E}_{\mathcal{D}, \mathcal{D}', \sigma}\left[\sup_{f \in \mathcal{V}, q \in \mathcal{Q}}\frac{1}{|\mathcal{D}|}\left|\sum_{x_i, y_i \in \mathcal{D}} \sigma_i\log f[x_i](y_i)q(\mathcal{E} = e_k|x_i, y_i)\right.\right. \tag{139}$$

$$\left.\left.-\sum_{x_i', y_i' \in \mathcal{D}'} \sigma_i\log f[x_i'](y_i')q(\mathcal{E} = e_k|x_i', y_i')\right|\right] \tag{140}$$

$$\leq \mathbb{E}_{\mathcal{D}, \sigma}\left[\sup_{f \in \mathcal{V}, q \in \mathcal{Q}}\frac{1}{|\mathcal{D}|}\left|\sum_{x_i, y_i \in \mathcal{D}} \sigma_i\log f[x_i](y_i)q(\mathcal{E} = e_k|x_i, y_i)\right|\right] \tag{141}$$

$$+ \mathbb{E}_{\mathcal{D}', \sigma}\left[\sup_{f \in \mathcal{V}, q \in \mathcal{Q}}\frac{1}{|\mathcal{D}'|}\left|\sum_{x_i', y_i' \in \mathcal{D}'} \sigma_i\log f[x_i'](y_i')q(\mathcal{E} = e_k|x_i', y_i')\right|\right] \tag{142}$$

$$= 2\mathscr{R}_{|\mathcal{D}|}(\mathcal{G}_{\mathcal{V}}^k), \tag{143}$$

where $\sigma_i$ are independent Rademacher variables. Equation 137 follows from Jensen's inequality and the convexity of sup. Equation 139 holds due to the symmetry of $\log f[x_i](y_i)q(\mathcal{E} = e_k|x_i, y_i) - \log f[x_i'](y_i')q(\mathcal{E} = e_k|x_i', y_i')$ and the argument that Radamacher variables preserve the expected sum of symmetric random variables with a convex mapping (Ledoux & Talagrand (1991), Lemma 6.3).

Substituting Equation 143 to Equation 126, we have for any $\delta \in (0, 1)$, with a probability over $1 - \delta$, $\forall f \in \mathcal{V}$, $\forall q \in \mathcal{Q}$, the following holds:

$$\left| q(\mathcal{E} = e_k)\mathbb{E}_q\left[-\log f[x](y)|\mathcal{E} = e_k\right] - \frac{1}{|\mathcal{D}|}\sum_{x_i, y_i \in \mathcal{D}} -\log f[x_i](y_i)q(\mathcal{E} = e_k|x_i, y_i)\right| \quad (144)$$

$$\leq 2\mathscr{R}_{|\mathcal{D}|}(\mathcal{G}_{\mathcal{V}}^k) + B\sqrt{\frac{2\log\frac{1}{\delta}}{|\mathcal{D}|}}. \quad (145)$$

Let $\tilde{f}_k = \arg\inf_f\{q(\mathcal{E} = e_k)\mathbb{E}_q\left[-\log f[x](y)|\mathcal{E} = e_k\right]\}$.

Let $\hat{f}_k = \arg\inf_f\{\frac{1}{|\mathcal{D}|}\sum_{x_i, y_i \in \mathcal{D}} -\log f[x_i](y_i)q(\mathcal{E} = e_k|x_i, y_i)\}$.

Now we have

$$q(\mathcal{E} = e_k)\mathbb{E}_q\left[-\log \tilde{f}_k[x](y)|\mathcal{E} = e_k\right] - \frac{1}{|\mathcal{D}|}\sum_{x_i, y_i \in \mathcal{D}} -\log \tilde{f}_k[x_i](y_i)q(\mathcal{E} = e_k|x_i, y_i) \quad (146)$$

$$\leq q(\mathcal{E} = e_k)H_{\mathcal{V}}(Y|X, \mathcal{E} = e_k) - \frac{1}{|\mathcal{D}|}\sum_{x_i, y_i \in \mathcal{D}} -\log \hat{f}_k[x_i](y_i)q(\mathcal{E} = e_k|x_i, y_i) \quad (147)$$

$$\leq q(\mathcal{E} = e_k)\mathbb{E}_q\left[-\log \hat{f}_k[x](y)|\mathcal{E} = e_k\right] - \frac{1}{|\mathcal{D}|}\sum_{x_i, y_i \in \mathcal{D}} -\log \hat{f}_k[x_i](y_i)q(\mathcal{E} = e_k|x_i, y_i). \quad (148)$$

Combining Equation 144 and Equation 146-148, the lemma is proved. $\square$

**Lemma 2.** *Assume $\forall x \in \mathcal{X}, \forall y \in \mathcal{Y}, \forall f \in \mathcal{V}$, $\log f[\emptyset](y) \in [-B, B]$ where $B > 0$. The definition of $\mathcal{G}_{\mathcal{V}}^k$ and $\mathscr{R}_N(\mathcal{G})$ follows from Lemma 1. Define $\hat{f}_k = \arg\inf_f \frac{1}{|\mathcal{D}|}\sum_{x_i, y_i \in \mathcal{D}} -\log f[\emptyset](y_i)q(\mathcal{E} = e_k|x_i, y_i)$.*

*Then for any $q \in \mathcal{Q}$, any $\delta \in (0, 1)$, with a probability over $1 - \delta$, we have*

$$\left| q(\mathcal{E} = e_k)H_{\mathcal{V}}(Y|\mathcal{E} = e_k) - \frac{1}{|\mathcal{D}|}\sum_{x_i, y_i \in \mathcal{D}} -\log \hat{f}_k[\emptyset](y_i)q(\mathcal{E} = e_k|x_i, y_i)\right| \quad (149)$$

$$\leq 2\mathscr{R}_{|\mathcal{D}|}(\mathcal{G}_{\mathcal{V}}^k) + B\sqrt{\frac{2\log\frac{1}{\delta}}{|\mathcal{D}|}}. \quad (150)$$

*Proof.* Similar to Lemma 1, we could prove that

$$\left| q(\mathcal{E} = e_k)H_{\mathcal{V}}(Y|\mathcal{E} = e_k) - \frac{1}{|\mathcal{D}|}\sum_{x_i, y_i \in \mathcal{D}} -\log \hat{f}_k[\emptyset](y_i)q(\mathcal{E} = e_k|x_i, y_i)\right| \quad (151)$$

$$\leq 2\mathscr{R}_{|\mathcal{D}|}(\mathcal{G}_{\mathcal{V}\emptyset}^k) + B\sqrt{\frac{2\log\frac{1}{\delta}}{|\mathcal{D}|}}, \quad (152)$$

where $\mathcal{G}_{\mathcal{V}\emptyset}^k = \{g|g(x, y) = \log f[\emptyset](y)q(\mathcal{E} = e_k|x, y), f \in \mathcal{V}, q \in \mathcal{Q}\}$.

According to the definition for the predictive family $\mathcal{V}$ (Xu et al. (2020), Definition 1), $\forall f \in \mathcal{V}$, there exists $f' \in \mathcal{V}$ such that $\forall x \in \mathcal{X}$, $f[\emptyset] = f'[x]$. Thus, $\mathcal{G}_{\mathcal{V}\emptyset}^k \subset \mathcal{G}_{\mathcal{V}}^k$, and therefore $\mathscr{R}_{|\mathcal{D}|}(\mathcal{G}_{\mathcal{V}\emptyset}^k) \leq \mathscr{R}_{|\mathcal{D}|}(\mathcal{G}_{\mathcal{V}}^k)$. Substituting into Equation 151, the lemma is proved. $\square$

**Lemma 3** ((Xu et al., 2020), Theorem 1). *Assume $\forall x \in \mathcal{X}, \forall y \in \mathcal{Y}, \forall f \in \mathcal{V}, \log f[x](y) \in [-B, B]$ where $B > 0$. Define a function class $\mathcal{G}_{\mathcal{V}}^* = \{g|g(x, y) = \log f[x](y), f \in \mathcal{V}\}$. The definition of $\mathscr{R}_N(\mathcal{G})$ follows from Lemma 1.*

*Then for any $\delta \in (0, 0.5)$, with a probability over $1 - 2\delta$, we have*

$$\left| \mathbb{I}_{\mathcal{V}}(X \to Y) - \hat{\mathbb{I}}_{\mathcal{V}}(X \to Y) \right| \le 4\mathscr{R}_{|\mathcal{D}|}(\mathcal{G}_{\mathcal{V}}^*) + 2B\sqrt{\frac{2\log\frac{1}{\delta}}{|\mathcal{D}|}}. \tag{153}$$

Finally we are prepared to prove Theorem 3.

*Proof of Theorem 3.* We first bound the error of empirical estimation with the sum of items in Lemma 1,2,3.

$$|\mathcal{H}_{\mathcal{V}}^{\mathscr{E}_K}(X \to Y) - \hat{H}_{\mathcal{V}}^{\mathscr{E}_K}(X \to Y; \mathcal{D})| \tag{154}$$

$$= \left| \left[ \sup_{\mathcal{E} \in \mathscr{E}_K} \mathbb{I}_{\mathcal{V}}(X \to Y|\mathcal{E}) - \mathbb{I}_{\mathcal{V}}(X \to Y) \right] - \left[ \sup_{\mathcal{E} \in \mathscr{E}_K} \hat{\mathbb{I}}_{\mathcal{V}}(X \to Y|\mathcal{E}; \mathcal{D}) - \hat{\mathbb{I}}_{\mathcal{V}}(X \to Y; \mathcal{D}) \right] \right| \tag{155}$$

$$\le \left| \sup_{\mathcal{E} \in \mathscr{E}_K} \mathbb{I}_{\mathcal{V}}(X \to Y|\mathcal{E}) - \sup_{\mathcal{E} \in \mathscr{E}_K} \hat{\mathbb{I}}_{\mathcal{V}}(X \to Y|\mathcal{E}; \mathcal{D}) \right| + \left| \mathbb{I}_{\mathcal{V}}(X \to Y) - \hat{\mathbb{I}}_{\mathcal{V}}(X \to Y; \mathcal{D}) \right| \tag{156}$$

$$\le \sup_{\mathcal{E} \in \mathscr{E}_K} \left| \mathbb{I}_{\mathcal{V}}(X \to Y|\mathcal{E}) - \hat{\mathbb{I}}_{\mathcal{V}}(X \to Y|\mathcal{E}; \mathcal{D}) \right| + \left| \mathbb{I}_{\mathcal{V}}(X \to Y) - \hat{\mathbb{I}}_{\mathcal{V}}(X \to Y; \mathcal{D}) \right| \tag{157}$$

$$= \sup_{q \in \mathcal{Q}} \left| \sum_{k=1}^K [q(\mathcal{E} = e_k)H_{\mathcal{V}}(Y|\mathcal{E} = e_k) - q(\mathcal{E} = e_k)H_{\mathcal{V}}(Y|X, \mathcal{E} = e_k)] \right. \tag{158}$$

$$\left. - \sum_{k=1}^K \left[ q(\mathcal{E} = e_k)\hat{H}_{\mathcal{V}}(Y|\mathcal{E} = e_k; \mathcal{D}) - q(\mathcal{E} = e_k)\hat{H}_{\mathcal{V}}(Y|X, \mathcal{E} = e_k; \mathcal{D}) \right] \right| \tag{159}$$

$$+ \left| \mathbb{I}_{\mathcal{V}}(X \to Y) - \hat{\mathbb{I}}_{\mathcal{V}}(X \to Y; \mathcal{D}) \right| \tag{160}$$

$$\le \sum_{k=1}^K \sup_{q \in \mathcal{Q}} \left| q(\mathcal{E} = e_k)H_{\mathcal{V}}(Y|\mathcal{E} = e_k) - q(\mathcal{E} = e_k)\hat{H}_{\mathcal{V}}(Y|\mathcal{E} = e_k; \mathcal{D}) \right| \tag{161}$$

$$+ \sum_{k=1}^K \sup_{q \in \mathcal{Q}} \left| q(\mathcal{E} = e_k)H_{\mathcal{V}}(Y|X, \mathcal{E} = e_k) - q(\mathcal{E} = e_k)\hat{H}_{\mathcal{V}}(Y|X, \mathcal{E} = e_k; \mathcal{D}) \right| \tag{162}$$

$$+ \left| \mathbb{I}_{\mathcal{V}}(X \to Y) - \hat{\mathbb{I}}_{\mathcal{V}}(X \to Y; \mathcal{D}) \right| \tag{163}$$

$$= \sum_{k=1}^K \sup_{q \in \mathcal{Q}} \left| q(\mathcal{E} = e_k)H_{\mathcal{V}}(Y|\mathcal{E} = e_k) - \frac{1}{|\mathcal{D}|} \sum_{x_i, y_i \in \mathcal{D}} -\log \hat{f}_k[x_i](y_i)q(\mathcal{E} = e_k|x_i, y_i) \right| \tag{164}$$

$$+ \sum_{k=1}^K \sup_{q \in \mathcal{Q}} \left| q(\mathcal{E} = e_k)H_{\mathcal{V}}(Y|X, \mathcal{E} = e_k) - \frac{1}{|\mathcal{D}|} \sum_{x_i, y_i \in \mathcal{D}} -\log \hat{f}_k'[\emptyset](y_i)q(\mathcal{E} = e_k|x_i, y_i) \right| \tag{165}$$

$$+ \left| \mathbb{I}_{\mathcal{V}}(X \to Y) - \hat{\mathbb{I}}_{\mathcal{V}}(X \to Y; \mathcal{D}) \right|, \tag{166}$$

where $\hat{f}_k = \arg\inf_f \frac{1}{|\mathcal{D}|} \sum_{x_i, y_i \in \mathcal{D}} -\log f[x_i](y_i)q(\mathcal{E} = e_k|x_i, y_i)$,

and $\hat{f}_k' = \arg\inf_f \frac{1}{|\mathcal{D}|} \sum_{x_i, y_i \in \mathcal{D}} -\log f[\emptyset](y_i)q(\mathcal{E} = e_k|x_i, y_i)$, for any $q \in \mathcal{Q}$ and $1 \le k \le K$.

For simplicity, let

$$\text{Err}_k = \sup_{q \in \mathcal{Q}} \left| q(\mathcal{E} = e_k) H_{\mathcal{V}}(Y|X, \mathcal{E} = e_k) - \frac{1}{|\mathcal{D}|} \sum_{x_i, y_i \in \mathcal{D}} -\log \hat{f}_k[x_i](y_i) q(\mathcal{E} = e_k | x_i, y_i) \right|. \tag{167}$$

$$\text{Err}'_k = \sup_{q \in \mathcal{Q}} \left| q(\mathcal{E} = e_k) H_{\mathcal{V}}(Y|X, \mathcal{E} = e_k) - \frac{1}{|\mathcal{D}|} \sum_{x_i, y_i \in \mathcal{D}} -\log \hat{f}'_k[\emptyset](y_i) q(\mathcal{E} = e_k | x_i, y_i) \right|. \tag{168}$$

$$\text{Err}^* = \left| \mathbb{I}_{\mathcal{V}}(X \to Y) - \hat{\mathbb{I}}_{\mathcal{V}}(X \to Y; \mathcal{D}) \right|. \tag{169}$$

Then, by Lemma 1,2,3,

$$\Pr\left[ |\mathcal{H}_K^{\mathcal{V}} - \hat{\mathcal{H}}_K^{\mathcal{V}}(\mathcal{D})| > 4(K+1)\mathscr{R}_{|\mathcal{D}|}(\mathcal{G}_{\mathcal{V}}) + 2(K+1)B\sqrt{\frac{2\log\frac{1}{\delta}}{|\mathcal{D}|}} \right] \tag{170}$$

$$\leq \Pr\left[ \sum_{i=1}^{K} \text{Err}_k + \sum_{i=1}^{K} \text{Err}'_k + \text{Err}^* > 4(K+1)\mathscr{R}_{|\mathcal{D}|}(\mathcal{G}_{\mathcal{V}}) + 2(K+1)B\sqrt{\frac{2\log\frac{1}{\delta}}{|\mathcal{D}|}} \right] \tag{171}$$

$$\leq \Pr\left[ \sum_{i=1}^{K} \text{Err}_k + \sum_{i=1}^{K} \text{Err}'_k + \text{Err}^* > \sum_{k=1}^{K} 4\mathscr{R}_{|\mathcal{D}|}(\mathcal{G}_{\mathcal{V}}^k) + 4\mathscr{R}_{|\mathcal{D}|}(\mathcal{G}_{\mathcal{V}}^*) + 2(K+1)B\sqrt{\frac{2\log\frac{1}{\delta}}{|\mathcal{D}|}} \right] \tag{172}$$

$$\leq \Pr\left[ \bigcup_{k=1}^{K} \left( \text{Err}_k > 2\mathscr{R}_{|\mathcal{D}|}(\mathcal{G}_{\mathcal{V}}^k) + B\sqrt{\frac{2\log\frac{1}{\delta}}{|\mathcal{D})|}} \right) + \bigcup_{k=1}^{K} \left( \text{Err}'_k > 2\mathscr{R}_{|\mathcal{D}|}(\mathcal{G}_{\mathcal{V}}^k) + B\sqrt{\frac{2\log\frac{1}{\delta}}{|\mathcal{D})|}} \right) \right. \tag{173}$$

$$\left. + \left( \text{Err}^* > 4\mathscr{R}_{|\mathcal{D}|}(\mathcal{G}_{\mathcal{V}}^*) + 2B\sqrt{\frac{2\log\frac{1}{\delta}}{|\mathcal{D}|}} \right) \right] \tag{174}$$

$$\leq \sum_{k=1}^{K} \Pr\left[ \text{Err}_k > 2\mathscr{R}_{|\mathcal{D}|}(\mathcal{G}_{\mathcal{V}}^k) + B\sqrt{\frac{2\log\frac{1}{\delta}}{|\mathcal{D})|}} \right] + \sum_{k=1}^{K} \Pr\left[ \text{Err}'_k > 2\mathscr{R}_{|\mathcal{D}|}(\mathcal{G}_{\mathcal{V}}^k) + B\sqrt{\frac{2\log\frac{1}{\delta}}{|\mathcal{D})|}} \right] \tag{175}$$

$$+ \Pr\left[ \text{Err}^* > 4\mathscr{R}_{|\mathcal{D}|}(\mathcal{G}_{\mathcal{V}}^*) + 2B\sqrt{\frac{2\log\frac{1}{\delta}}{|\mathcal{D}|}} \right] \tag{176}$$

$$\leq 2(K+1)\delta. \tag{177}$$

Equation 172 is because of $\mathcal{G}_{\mathcal{V}}^k = \mathcal{G}_{\mathcal{V}}, \mathcal{G}_{\mathcal{V}}^* \subset \mathcal{G}_{\mathcal{V}}$ and therefore $R_{|\mathcal{D}|}(\mathcal{G}_{\mathcal{V}}^k) \leq R_{|\mathcal{D}|}(\mathcal{G}_{\mathcal{V}}), R_{|\mathcal{D}|}(\mathcal{G}_{\mathcal{V}}^*) \leq R_{|\mathcal{D}|}(\mathcal{G}_{\mathcal{V}})$.

Hence,

$$\Pr\left[ |\mathcal{H}_{\mathcal{V}}^{\mathcal{E}_K}(X \to Y) - \hat{H}_{\mathcal{V}}^{\mathcal{E}_K}(X \to Y; \mathcal{D})| \leq 4(K+1)\mathscr{R}_{|\mathcal{D}|}(\mathcal{G}_{\mathcal{V}}) + 2(K+1)B\sqrt{\frac{2\log\frac{1}{\delta}}{|\mathcal{D}|}} \right] \tag{178}$$

$$\geq 1 - 2(K+1)\delta. \tag{179}$$

$\square$

## H PROOF OF THEOREM 4

*Proof of Theorem 4.* The objective function of our IM algorithm is directly derived from the definition of empirical predictive heterogeneity in Definition 6. For the regression task, we assume the predictive family as

$$\mathcal{V}_1 = \{g : g[x] = \mathcal{N}(f_\theta(x), \sigma^2), f \text{ is the regression model and } \theta \text{ is learnable}, \sigma = 1.0(\text{fixed})\}, \quad (180)$$

where we only care about the output of the model and the noise scale of the Gaussian distribution is often ignored, for which we simply set $\sigma = 1.0$ as a fixed term. Then for each environment $e \in \text{supp}(\mathcal{E}^*)$, the $\mathbb{I}_\mathcal{V}(X \to Y | \mathcal{E}^* = e)$ becomes

$$\mathbb{I}_\mathcal{V}(X \to Y | \mathcal{E}^* = e) \propto \min_\theta \mathbb{E}^[\|Y - f_\theta(X)\|^2 | \mathcal{E}^* = e] - \text{Var}(Y | \mathcal{E}^*), \quad (181)$$

which corresponds with the MSE loss and the proposed regularizer in Equation 17. For the classification task, the derivation is similar, and the regularizer becomes the entropy of $Y$ in sub-population $e$ and the loss function becomes the cross-entropy loss. □

## I DISCUSSION ON DIFFERENCES WITH SUB-GROUP DISCOVERY

Subgroup discovery (SD, (Helal, 2016)) is aimed at extracting "interesting" relations among different variables ($X$) with respect to a target variable $Y$. Coverage and precision of each discovered group is the focus of such method. To be specific, it learns a partition on $P(X)$ such that some target label $y$ dominates within each group. The most siginficant gap between subgroup discovery and our predictive heterogeneity lies in the pattern of distributional shift among clusters: for subgroup discovery, $P(X)$ and $P(Y)$ varies across subgroups but there is a universal $P(Y|X)$. While for predictive heterogeneity $P(Y|X)$ differs across sub-population, which indicates diversified prediction mechanism. It is such disparity of prediction mechanism that inhibits the performance of a universal predictive model on a heterogeneous dataset, which is the emphasis of OOD problem and group fairness.

We think sub-group discovery is more applicable for settings where the distributional shift is minor while high explainability is required, since it generates simplified rules that people can understand. Also, sub-group discovery methods is suitable for the settings that only involve tabular data (typically from a relational database), where the input features have clear semantics. And our proposed method could deal with general machine learning settings, including complicated data (e.g., image data) that involves representation learning. Also, when people have to handle settings where data heterogeneity w.r.t. prediciton mechanism exists inside data, our method is more applicable. However, both kinds of methods can be used to help people understand data and make more reasonable decisions.

## J DISCUSSION ON THE POTENTIAL FOR FAIRNESS

We find combining our measure with algorithmic fairness is an interesting and promising direction and we think our measure has the potential to deal with algorithmic bias. Our method could generate sub-populations with possibly different prediction mechanisms, which could do some help in the following aspects:

**Risk feature selection**: we could select features according to our predictive heterogeneity measure to see what features bring the largest heterogeneity. If they are sensitive features, people should avoid their effects, and if they are not, they could direct people to build better machine learning models.

**Examine the algorithmic fairness**: we could use the learned sub-populations to examine whether a given algorithm is fair by calculating the performance gap across the sub-populations.

