# OpenReview forum: "Measure the Predictive Heterogeneity"
_ICLR.cc/2023/Conference — ICLR 2023 poster_

### Official Review · Reviewer_QrpC · 2022-10-23

**Confidence:** 3
**Correctness:** 3
**Technical Novelty And Significance:** 2
**Empirical Novelty And Significance:** 2
**Recommendation:** 6

**Clarity, Quality, Novelty And Reproducibility:**

Clarity: Good. The paper is well organised, and easy to follow.
Quality: I have not spotted a flaw.
Novelty: moderate. The empirical impact is low.
Reproducibility: Most details are provided for reproducibility.


**Strength And Weaknesses:**

S1: The paper is well written and easy to follow.

S2: The theoretical analysis is substantial and looks sound.

S3: The experimental case studies are from different areas.


W1: The predictive heterogeneity criterion is interesting, but is incremental to the previous work (Xu et al.). In Definition 5, $\Epsilon$ is unknown, and the authors' solutions for finding $\Epsilon$  are in linear settings with independent noise. These assumptions will restrict the empirical impact of the work.  Also, since $\Epsilon$ is unknown, setting a $k$ is challenging and the authors have not discussed how to set a $k$.

W2: The empirical impact of the proposed method has not been shown to be significant. In three experimental demonstrations in subpopulation division, the number of k is 2 (the minimum possible number in heterogeneity discovery). Finding data heterogeneity is not a new topic and many works have been done. For example, in cancer subtype discovery in bioinformatics, many sophisticated methods have been proposed to deal with complex data types for many subgroups. The paper lacks discussions and comparisons with this type of work.

====

The authors' detailed explanations and follow-up experiments clarify my doubts.

**Summary Of The Paper:**

The paper studies criteria and methods for finding data heterogeneities that affect building predictive models. The authors define a predictive homogeneity criterion and present an algorithm to find predictive heterogeneity in data. The authors further show predictive heterogeneity criterion can provide insights for subpopulation in agriculture, sociology and object recognition applications and its benefits for OOD generalisation.

**Summary Of The Review:**

It is a technically sound paper with a modest empirical impact paper. I am not excited by the idea or the experimental results.

---

> ### Author Response · Authors · 2022-11-09
> **To Reviewer QrpC**
>
> We sincerely appreciate your approval of the theoretical values of this work. Thank you for the suggestions. We would like to address your concerns proposed in your review, which we have summarized into 5 questions.
>
> ### **Q1: The novelty of this work**
> We want to clarify the novelty of this work compared with the previous work [1].
>
> **Firstly**, we address the important problem of data heterogeneity and propose the **first formulation of predictive heterogeneity**, which quantifies the maximal additional usable information that can be gained by generating sub-populations. Based on the newly-proposed measure, we demonstrate its **theoretical properties (Proposition 1, Theorem 1, Theorem 2)** and prove that it can be estimated **under finite samples (Theorem 3)**.
> And we also derive an empirical algorithm to estimate it.
>
> **Secondly**, the target of this work is different from [1]. This work aims to quantify the predictive heterogeneity inside data, while [1] proposes to define the term usable information and does not consider the data heterogeneity problem.
>
> **Thirdly**, from the experiments, we show that some typical OOD generalization algorithms could gain benefits from our learned sub-populations, which also validates the effectiveness of the proposed measure and algorithm.
>
>
> [1] Xu, Y., Zhao, S., Song, J., Stewart, R., & Ermon, S. (2020). A theory of usable information under computational constraints. ICLR.
>
> ### **Q2:  Solutions for finding $\mathcal E$ are in linear settings with independent noise**
> Our solutions for finding $\mathcal E$ are designed for **general regression and classification tasks**, and we **do not require** the linear settings and independent noises.
>
> * In Section 4, we propose the IM algorithm to estimate the measure as well as find sub-populations $\mathcal E$ for general machine learning tasks. Theorem 4 demonstrates the equivalence between our objective function (Equation 15) and the empirical predictive heterogeneity.
>
> * In Section 5, empirically, we use MLP for the agriculture example, ResNet-18 for the object recognition example, and MLP for the colored MNIST experiment.
> Therefore, our proposed algorithm is general and could deal with complicated data.
>
> * In Section 3.2, we demonstrate the general properties of predictive heterogeneity.
> And in Section 3.3, we demonstrate the theoretical properties under linear cases to allow readers to gain an intuitive association with the properties of the methods and deliberate on their applicability (*as agreed by Reviewer E69D*).
> However, it does not mean that our algorithm can only be used under linear cases.
>
>
>
> ### **Q3: Different $K$s**
>
> In Appendix B, we give the results of choosing different $K$s for the simulated experiment in Section 5.2.
> Also, we add one more experiment to show that (1) when the chosen $K$ is smaller than the ground truth, the performances of our methods will drop but are still better than ERM (2) when the chosen $K$ is larger, the performances are not affected much (consistent with the results in Appendix B).
>
> Experiment Setting: The input features $X=[S,T,V]\in\mathbb{R}^{10}$ consist of stable features $S\in\mathbb{R}^5$, noisy features $T\in\mathbb{R}^4$ and the spurious feature $V\in\mathbb R$:
> $$
> S\sim \mathcal{N}(2,2\mathbb I_5),\quad  T\sim \mathcal{N}(0, 2\mathbb I_4), \quad Y=\theta_S^TS + S_1S_2S_3+\mathcal{N}(0,0.5),
> $$
> and we generate the spurious feature via the:
> $$
> V = \theta_V^e Y + \mathcal{N}(0, 0.3),
> $$
> where $\theta_V^e$ varies across sub-populations and is dependent on which sub-population the data point belongs to.
> In training, we sample 8000 data points from $e_1$ with $\theta_V^1=3.0$, 1000 points from $e_2$ with $\theta_V^2=-1.0$, 1000 points from $e_3$ with $\theta_V^3=-2.0$ and 1000 points from $e_4$ with $\theta_V^4=-3.0$.
> Therefore, the ground-truth number of sub-populations is 4.
> In testing, we test the performances on $e_4$ with $\theta_V^4=-3.0$, which has strong distributional shifts from training data.
> The average MSE over 10 runs are:
>
> | Method       | $K=2$ | $K=3$ | $K=4$ | $K=5$ | $K=6$ | ERM   |
> |--------------|-------|-------|-------|-------|-------|-------|
> | Ours+Balance | 0.745 | 0.656 | 0.632 | 0.643 | 0.645 | 0.813 |
> | Ours+IRM     | 0.668 | 0.613 | 0.595 | 0.588 | 0.606 | 0.813 |
> | Ours+IGA     | 0.681 | 0.651 | 0.603 | 0.577 | 0.597 | 0.813 |
>
> From the results, we can see that when $K$ is smaller than the ground-truth, increasing $K$ benefits the OOD generalization performance, and when $K$ is larger, the performances are not affected much, which is consistent with the results in Appendix B.

---

> > ### Author Response · Authors · 2022-11-09
> > **Q4: How to choose $K$.**
> >
> > We make some discussions on how to choose $K$ and add them to Appendix B.
> >
> > **Firstly**, choosing an appropriate cluster number $K$ is a common but important problem in the literature of clustering.
> > For our IM algorithm, we think there are mainly two ways to choose $K$:
> >
> > * According to the predictive heterogeneity index: When the chosen $K$ is smaller than the ground truth, our measure tends to increase quickly when increasing $K$; and when $K$ is larger than the ground truth, the increasing speed will slow down, which could direct people to choose an appropriate $K$.
> > * According to the prediction model: Since our IM algorithm aims to learn sub-populations with different prediction mechanisms, one could compare the learned model parameters $\theta_1, \dots, \theta_K$ to judge whether $K$ is much larger than the ground truth, i.e., if two resultant models are quite similar, $K$ may be too large (some learned sub-populations may from the same ground-truth one). For linear models, one can directly compare the coefficients. For deep models, we think one can calculate the transfer losses across sub-populations.
> >
> >
> > **Secondly**, when our algorithm serves for downstream tasks (e.g., the out-of-distribution generalization task in Section 5.2), the models' performances are not quite sensitive to the choice of $K$s, as shown in our added experiments above and Appendix B. That is, if we set $K$ in an appropriate range, the performances on the downstream tasks can gain some benefits from us.
> >
> > ### **Q5: Discussions and comparisons with subtype discovery.**
> > We add some discussions on the literature of the subtype discovery in Bioinformatics, especially for cancer diagnosis.
> >
> > Cancer subtype discovery is typically handled by unsupervised clustering or supervised classification [1,5], which are both essentially distinct from our predictive heterogeneity discovery in terms of fundamental targets.
> >
> > **Firstly**, graph clustering is adopted for cancer subtype discovery [5], which recognizes cancer types from a graph constructed by RNA expression profiles (feature $X$). Such methods are essentially clustering algorithms on $P(X)$. Supervised classification is another path toward cancer subtype discovery. For example, in [2,3] relevant features are extracted by implementing the SD subgroup discovery algorithm for the detection of different cancer types, where SD is a heuristic beam search-based algorithm in which the final subgroup set is selected according to the opinion of an expert instead of simply using a measure for subgroup searching and selecting. And [4] adopts an SD approach SD4TS (Subgroup Discovery for Test Selection) for breast cancer diagnosis. Another classification-based method involves ANN [5]. Their target is to predict cancer type $Y$ with gene expression matrix $X$ by fitting $P(Y|X)$. Since the learned sub-group is expected to have the same class label, the data distribution differs in $P(X)$ or $P(Y)$ across subgroups, but $P(Y|X)$ is invariant.
> >
> > **Secondly**, for predictive heterogeneity, **$P_e(Y|X)$ differs across sub-populations**, which indicates diversified prediction mechanism. It is a such disparity of prediction mechanism that inhibits the performance of a universal predictive model on a heterogeneous dataset, which is the emphasis of the OOD generalization problem and group fairness.
> >
> > **Further**, despite that specific algorithms have been devised for cancer subtype discovery, our proposed method **could deal with general machine learning tasks and is compatible with any predictive models** including deep neural networks, which is much more general.
> >
> >
> >
> > [1] Helal, S. (2016). Subgroup discovery algorithms: a survey and empirical evaluation. Journal of computer science and technology, 31(3), 561-576.
> >
> > [2] Carmona, C. J., González, P., del Jesus, M. J., Navío-Acosta, M., & Jiménez-Trevino, L. (2011). Evolutionary fuzzy rule extraction for subgroup discovery in a psychiatric emergency department. Soft Computing, 15(12), 2435-2448.
> >
> > [3] Mueller M, Rosales R, Steck H, Krishnan S, Rao B, Kramer S. Subgroup discovery for test selection: A novel approach and its application to breast cancer diagnosis. In Proc. the 8th Intelligent Data Analysis, Aug.31-Sept.2, 2009, pp.119- 130.
> >
> > [4] Trajkovski, I., Zelezny, F., Lavrac, N., & Tolar, J. (2007). Learning relational descriptions of differentially expressed gene groups. IEEE Transactions on Systems, Man, and Cybernetics, Part C (Applications and Reviews), 38(1), 16-25.
> >
> > [5] Vasudevan, P., & Murugesan, T. (2018). Cancer subtype discovery using prognosis-enhanced neural network classifier in multigenomic data. *Technology in cancer research & treatment*.
> >
> > **We welcome further technical advices for this work, and we're ready to address your concerns.**

---

> > > ### Comment · Reviewer_QrpC · 2022-11-19
> > > **Thanks for the explanations and revision**
> > >
> > > Thank you for the detailed explanations and follow-up experiments and revision. I will update my review.

---

> > > > ### Author Response · Authors · 2022-11-20
> > > > **Thanks for your support**
> > > >
> > > > Thanks for your efforts and support for this work!

---

### Official Review · Reviewer_E69D · 2022-10-24

**Confidence:** 4
**Correctness:** 3
**Technical Novelty And Significance:** 4
**Empirical Novelty And Significance:** 3
**Recommendation:** 6

**Clarity, Quality, Novelty And Reproducibility:**

The proposed method is novel and at the same time well motivated from existing literature. The presentation by itself is ok but could be improved upon. For improved reproducibility, distributing their code in open source may be beneficial

**Strength And Weaknesses:**

There are several strong aspects of the paper as follows

- The authors proposed a formal definition of predictive heterogeneity by starting from the first principles of probablity and using concepts from information theory. This allows the authors to immediately connect certain strong theoretical results for the proposed definition
- The theoretical analysis around linear cases is interesting. While not being the most real-world scenario, this analysis allows readers to gain an intuitive association with the properties of the methods and deliberate on its applicability
- The empirical exploration around the yield datasets is motivating. Similarly the other empirical evaluations lends substance to the otherwise theoretical paper

The paper could still be improved upon certain manner
- The presentation of the paper could be improved upon. While the background around information theory is useful, more intuitions around the theoretical propositions could improve the readability of the paper
- While acknowledging that the paper is theoretical, the empirical results do not sufficiently motivate the "correctness" of the method. The OOD experiments are interesting, however, It may be useful to devise some sort of quantitative/qualitative measure of extracted sub-groups from the 3 example tasks. There is a huge body of literature around sub-group discovery that can provide pointers for this (e.g using a synthetic dataset)
- The applicability of the algorithm is also not well discussed. There are many sub-group discovery methods. Are there situations where other methods are more applicable? Can you provide more details on the computational complexity

**Summary Of The Paper:**

The authors analyzed the problem of identifying differing prediction mechanisms amongst different sub-populations that are likely to differ. They provided theoretical outlining of the mechanism and proposed a novel algorithm to explore the differences. The provided empirical analysis of the explore differences and provided discussions around the usefulness of extracted insights

**Summary Of The Review:**

Overall the paper is well motivated, has strong theoretical under-pinnings, and backed by some level of empirical evidence. While the empirical part of the paper may be improved upon, the proposed method could be an interesting contribution, especially when considering the problems of algorithmic bias and model selection.
Apart from the aforementioned aspects, below are a few other aspects that the authors may consider

- Provide a glossary of symbols used for the theoretical descriptions. Define symbols before using them (e.g. Section 2, Notations the probability triplet is not described)
- Provide more details around the complexity of the proposed optimization. As the authors have noted, MI is notoriously intractable - providing more details on the final complexity would be useful for consideration of the algorithm to a wide range of problems
- Considering expanding the discussions around the benefits to quantifying/debiasing w.r.t algorithmic fairness of models

Edit: The authors have addressed a number of concerns regarding validity in their response

---

> ### Author Response · Authors · 2022-11-09
> **To Reviewer E69D**
>
> We sincerely appreciate your approval of the idea, the novelty of our methods, and the theoretical analysis of this work. Thank you for the suggestions. We revise the paper according to your suggestions and would like to address your concerns proposed in your review, which we have summarized into 4 questions.
>
> ### **Q1:Quantitative/qualitative measure of extracted sub-groups**
> Devising the quantitative measure of the learned sub-population of our method is an interesting problem.
>
> **Firstly**, the proposed predictive heterogeneity itself is a quantitative measure to characterize how much additional usable information can be gained by exploring the sub-population.
> And during the estimation of the predictive heterogeneity, our IM algorithm learns the sub-populations to maximize the usable information.
> Therefore, the learned sub-populations have maximal additional information that benefits prediction.
> Further, to demonstrate the precision of our IM algorithm, in Appendix D, we **visualize** the **estimated empirical predictive heterogeneity** by our IM algorithm, the theoretical approximations as well as the empirical approximations under finite samples.
> From Figure 7, under the homogeneous case (no heterogeneity), our estimated heterogeneity approaches 0, and under heterogeneous cases, we can see that the learned sub-populations have high predictive heterogeneity, which could reflect that our IM algorithm has good precision.
>
>
> **Secondly**, to evaluate the 'correctness' of learned sub-populations since our predictive heterogeneity serves for prediction, we think it reasonable to use the prediction performances for further evaluation.
> Therefore, in Section 5.2, we test the out-of-distribution generalization performances with typical OOD generalization methods plus our learned sub-populations.
> From the OOD generalization performances, three typical methods (Balance, IRM, IGA) could all achieve good performance with our learned sub-populations, which shows that our learned sub-populations are generally helpful for OOD generalization.
> Also, we also test the performances of these three methods combined with Kmeans, whose performances drop a lot.
> And this shows the rationality of our learned sub-populations.
>
> **Thirdly**, from the sub-group discovery literature [1], subgroup discovery is to mine some rules w.r.t. a special property of interest, which is similar to a path in the decision tree.
> And the learned sub-group is expected to have the same class label.
> Correspondingly, the measures in sub-group discovery literatures mainly focus on the coverage (e.g., AvCov, AvSup), precision (e.g., AvConf), significance (e.g., AvSig) and quality (e.g., AvWRAcc) of the discovered rules that split the dataset into sub-groups. These measures put an emphasis on the purity of labels such that the label distribution $P_e(Y)$ within each group $e$ should be maximally diverged from the overall label distribution $P(Y)$.
> However, our predictive heterogeneity measures the maximal additional information that is useful for prediction. Thus, the divergence between $P_e(Y|X)$ and $P(Y|X)$ is maximized but we hope the labels are **balanced** in each learned sub-population (the effect of regularizer $U_{\mathcal V}(W,Y_N)$).
> Therefore, the measures in sub-group discovery cannot be directly applied here.
>
> **Then**, corresponding with your suggestions, we directly **calculate the 'correctness' of the learned sub-population in the simulation experiment** in Section 5.2. Since in this experiment the ground-truth sub-population can be obtained, the 'correctness' measure could be the accuracy of discovered group label (allowing a permutation of group indices).
>
> | KMeans 	| EIIL   	| Ours   	|
> |----|----|----|
> | 0.5264 	| 0.5137 	| 0.8863 	|
>
> From the results, we can see that our IM algorithm could obtain the desired sub-populations.
>
>
>
>
> [1] Helal, S. (2016). Subgroup discovery algorithms: a survey and empirical evaluation. Journal of computer science and technology, 31(3), 561-576.

---

> > ### Author Response · Authors · 2022-11-09
> > **Q2: Complexity of the proposed optimization**
> >
> > For the optimization, we use truncated back-propagation [1].
> > According to [1], denote the parameter dimension as $M$ and sample size as $N$, and the additional time complexity per run is $\mathcal O(tMNK)$.
> > * $t$ is the number of steps of the truncated back-propagation, which is usually very small since the bias can be exponentially small in $t$. And in our experiments, $t=1$ already gives satisfying results.
> > * $K$ is the pre-defined number of sub-populations, which is often small (typically we have $K<10$).
> > * This additional time complexity is **acceptable** because the time complexity of one-time forward is $\mathcal O(DMN)$, where $D$ is the sample dimension and is usually large (e.g. for image data $D\gg tK$).
> > * Further, the approximation process in the truncated back-propagation is **GPU-friendly**, which supports GPU and could be significantly accelerated. For example, [2] and [3] use the same bi-level technique on **deep neural networks (ResNet-18 and ResNet-32)**.
> >
> >
> > [1] Shaban, A., Cheng, C. A., Hatch, N., & Boots, B. (2019, April). Truncated back-propagation for bilevel optimization. In The 22nd International Conference on Artificial Intelligence and Statistics (pp. 1723-1732). PMLR.
> >
> > [2] Zhou, X., Lin, Y., Pi, R., Zhang, W., Xu, R., Cui, P., \& Zhang, T. Model agnostic sample reweighting for out-of-distribution learning. In International Conference on Machine Learning (pp. 27203-27221). PMLR.
> >
> > [3] Shu, J., Xie, Q., Yi, L., Zhao, Q., Zhou, S., Xu, Z., & Meng, D. (2019). Meta-weight-net: Learning an explicit mapping for sample weighting. Advances in neural information processing systems, 32.
> >
> >
> >
> > ### **Q3: Applicability of the algorithm**
> > Thanks for your constructive suggestion.
> > We make a discussion on this and add it to Appendix K in the revised version.
> >
> > Subgroup discovery (SD) is aimed at extracting "interesting" relations among different variables ($X$) with respect to a target variable $Y$. [1]  The coverage and precision of each discovered group are the focus of such methods. To be specific, it learns a partition on $P(X)$ such that some target label $y$ dominates within each group. The most significant gap between subgroup discovery and our predictive heterogeneity lies in the pattern of the distributional shift among clusters: for subgroup discovery, $P(X)$ and $P(Y)$ vary across subgroups but there is a universal $P(Y|X)$. While for predictive heterogeneity $P(Y|X)$ differs across sub-population, which indicates a diversified prediction mechanism. It is a such disparity of prediction mechanism that inhibits the performance of a universal predictive model on a heterogeneous dataset, which is the emphasis of the OOD generalization problem and group fairness.
> >
> > We think sub-group discovery is more applicable for settings where the **distributional shift is minor** while **high explainability is required** since it generates simplified rules that people can understand. Also, sub-group discovery methods are suitable for settings that only involve tabular data (typically from a relational database), where the input features have clear semantics.
> > And our proposed method could deal with **general machine learning settings**, including complicated data (e.g., image data) that involves representation learning.
> > Also, when people have to handle settings where **data heterogeneity w.r.t. prediction mechanism** exists inside data, our method is more applicable.
> > However, **both kinds of methods can be used to help people understand data and make more reasonable decisions**.
> >
> >
> > [1] Helal, S. (2016). Subgroup discovery algorithms: a survey and empirical evaluation. *Journal of computer science and technology*, *31*(3), 561-576.
> >
> > ### **Q4: Discussions around the benefits to quantifying/debiasing w.r.t algorithmic fairness of models.**
> > Thanks for your constructive suggestion, and we make some discussions here and add them to Appendix L in the revised version.
> >
> > We find combining our measure with algorithmic fairness is an interesting and promising direction and we think our measure has the potential to deal with algorithmic bias.
> > Our method could generate sub-populations with possibly different prediction mechanisms, which could help in the following aspects:
> >
> > *  Risk feature selection: we could select features according to our predictive heterogeneity measure to see what features bring the largest heterogeneity. If they are sensitive features, people should avoid their effects, and if they are not, they could direct people to build better machine learning models.
> > *  Examine the algorithmic fairness: we could use the learned sub-populations to examine whether a given algorithm is fair by calculating the performance gap across the sub-populations.

---

> ### Author Response · Authors · 2022-11-20
> **Further Reply to Reviewer E69D**
>
> We sincerely appreciate your approval and suggestions to improve this paper and we would really like to know if you have any further concerns or ambiguity regarding this paper. We hope to make the most of this opportunity to shed light on this paper as until now, **all the reviewers support the acceptance of this work** after our rebuttal. We believe this paper will contribute to the machine-learning community and many sociological fields.
> Therefore, we want to **make all our efforts to make sure that all your concerns could be well-addressed**.
>
>
> According to your suggestions, we summarize our revisions as (some of them are not placed in the main body due to space limits, and we promise to add them in the camera-ready version)
>
> * **Notations**: For a probability triple $(\mathbb S,  \mathcal F, \mathbb P)$, where $\mathbb S$ denotes the sample space, $\mathcal F$ denotes the event space and $\mathbb P$ denotes a probability function, define random variables $X: \mathbb S\rightarrow \mathcal X$ and $Y: \mathbb S\rightarrow \mathcal Y$ where $\mathcal X$ is the covariate space and $\mathcal Y$ is the target space. Accordingly. $x \in \mathcal X$ denotes the covariates, and $y\in\mathcal{Y}$ denotes the target. Denote the set of random categorical variables as $\mathcal C = \{ C: \mathbb S \rightarrow \mathbb N| \text{supp}(C) \;\text{is finite} \}$. $\mathscr E \subseteq \mathcal C$ denotes an environment set.
> Additionally, $\mathcal{P}(\mathcal{X}), \mathcal{P}(\mathcal Y)$ denote the set of all probability measures over the Borel algebra on the spaces $\mathcal{X}, \mathcal{Y}$ respectively.
> $H(\cdot)$ denotes the Shannon entropy of a random variable, and $H(\cdot|\cdot)$ denotes the conditional entropy of two random variables.
> * **Complexity of the algorithm**: we analyze the extra computational complexity of our method to show that the additional complexity is acceptable and friendly to deep models.
> * **Discussions around the benefits of quantifying/debiasing w.r.t algorithmic fairness of models**: we add the discussion on the potential usage for fairness in Appendix L.
> * **Quantitative/qualitative measure of extracted sub-groups**: we add the clustering accuracy in the simulated experiments and clarify the difference between our methods and some sub-group discovery methods in detail in Appendix K.
>
> **We welcome any further technical advice or questions on this work and we will make our best to address your concerns.**

---

### Official Review · Reviewer_uqaW · 2022-10-31

**Confidence:** 2
**Correctness:** 4
**Technical Novelty And Significance:** 3
**Empirical Novelty And Significance:** 3
**Recommendation:** 8

**Clarity, Quality, Novelty And Reproducibility:**

I find the paper fast-paced. The authors seem to be trying to compactly include information which may get overwhelming to a person not familiar with the literature. Moreover, the paper concerns itself with definitions and intricacies from the get-go which is expected from a mathematically rigorous paper but on the other hand fails to provide the overarching flow. In other words, it loses the forest for the trees.

**Strength And Weaknesses:**

The paper seems interesting and timely. Its implications are general and the experiments show its usefulness in providing insights as well as OOD generalization.

**Summary Of The Paper:**

This work defines predictive heterogeneity to quantify the heterogeneity in the data which influences the predictive performance. It proposes an algorithm which quantifies the mentioned heterogeneity and use that in examples to show how it can be helpful in understanding the subtleties of the data.

**Summary Of The Review:**

This work is significant and it has very good applicability.

---

> ### Author Response · Authors · 2022-11-09
> **To Reviewer uqaW**
>
> We sincerely appreciate your approval of the idea and the novelty of this work and thank you for the suggestions on the presentations. We would like to add more background knowledge as well as more intuitions to make it easier to follow.

---

### Official Review · Reviewer_6bwF · 2022-11-04

**Confidence:** 4
**Correctness:** 3
**Technical Novelty And Significance:** 3
**Empirical Novelty And Significance:** 4
**Recommendation:** 6

**Clarity, Quality, Novelty And Reproducibility:**

This paper clearly presents the crucial idea of measuring data heterogeneity using predictive heterogeneity. However, some technical details are not well explained.

**Strength And Weaknesses:**

Strengths
(1) It introduces a predictive heterogeneity to define the data heterogeneity quantitatively.
(2) It theoretically analyzes how predictive heterogeneity works in homogenous and heterogeneous settings, and studies its empirical estimate with PAC bounds.
(3) The experiments demonstrate the effectiveness of the proposed Information Maximization algorithm in various prediction tasks.

Weaknesses
(1) One major concern is the tightness of the predictive heterogeneity in Theorem 2. The first term Var[r]/E[r^2] involves both the variance of r and the expectation of r^2. It is unclear how the E[r^2] affects the approximation tightness.
(2) In the introduction, it mentioned the data heterogeneity in the graph data. Could the proposed predictive heterogeneity be applied to learn the data heterogeneity of graph data? If so, how is the node interdependence reflected in the predictive heterogeneity?
(3) It mentioned that "Large H(P) indicates that the correlation between X and Y is enhanced by E" in Section 3.1. What is H(P) here? Why does it indicate the correlation between X and Y?
(4) How is the regularize U_V(W, Y_N) defined in the objective function of IM?
(5) Theorem 4 shows the connection between the optimization of Eq. (15) and the empirical predictive heterogeneity. Would this result consider the approximation of Eq. (20-21)?
(6) Appendix B showed that the proposed method is not sensitive to the choices of K. But K might be much smaller than the real number of sub-populations. In this case, the model performance might be improved by increasing the value of K.
(7) In Table 1, it is unclear why EIL (Creager et al., 2021) fails to predict the simulated data.

###########
Update: The authors have addressed most of my concerns, and thus I would like to improve the score accordingly.

**Summary Of The Paper:**

This paper studied the problem of measuring the data heterogeneity, i.e., the distributional diversity inside the data. It provided a novel data heterogeneity notion named predictive heterogeneity to explore how it affects the prediction of machine learning models. The crucial idea was to define the environment-conditional predictive V-information, which enabled the empirical estimation from finite samples. Its efficacy was then verified in various prediction tasks.

**Summary Of The Review:**

The overall idea of this paper is interesting. It theoretically and empirically shows the quality of predictive heterogeneity in measuring data heterogeneity. I would like to improve my score if my concerns can be solved well.

---

> ### Author Response · Authors · 2022-11-09
> **To Reviewer 6bwF**
>
> We sincerely appreciate your approval of the idea and the theoretical analysis of this work and thank you for the suggestions. We revise the paper according to your suggestions and would like to address your concerns proposed in your review, which we have summarized into 7 questions.
>
> ### **Q1. The tightness of the approximation error bound in Theorem 2**
>
> Thanks for your constructive suggestions.
> In the previous version of Theorem 2, we do not consider the noise scale on the spurious  $V$ during the derivation.
> And here we revise Theorem 2 and give a more precise analysis of this setting.
>
> The revised Theorem 2 is:
>
> For the prediction task, $X=[S,V]^T\rightarrow Y$ with a latent environment variable $\mathcal E^*$, the data generation process with selection bias is defined as:
>
> $$
>     Y = \beta S + f(S) + \epsilon_Y, \epsilon_Y \sim \mathcal{N}(0, \sigma_Y^2); \quad V = r(\mathcal E^*) f(S) + \sigma(\mathcal E^*)\cdot\epsilon_V, \epsilon_V \sim \mathcal{N}(0, 1),
> $$
>
> where  $f:\mathbb R\rightarrow \mathbb R$ and $r,\sigma:\text{supp}(\mathcal E^*) \rightarrow \mathbb R$ are measurable functions. $\beta \in \mathbb R$.
> Assume that $\mathbb{E}[f(S)S]=0$ and there exists $L>1$ such that $L\sigma^2(\mathcal E^*)< r^2(\mathcal E^*)\mathbb{E}[f^2]$. For the predictive family defined in equation 11 and the environment set $\mathscr E = \mathcal C$, the predictive heterogeneity of the prediction task $[S,V]^T\rightarrow Y$ approximates to:
>
> $$
> 		\mathcal{H}^\mathcal C_{\mathcal{V}}(X\rightarrow Y) \approx \frac{1}{2}\frac{\text{Var}(r_e)\mathbb{E}[f^2]+\mathbb{E}[\sigma^2(\mathcal E^*)]}{\mathbb{E}[r_e^2]\mathbb{E}[f^2]+\mathbb{E}[\sigma^2(\mathcal E^*)]}\mathbb{E}[f^2(S)],
> $$
> and the approximation error bound is
>
> $$
> \frac{1}{2}\max(\sigma_Y^2,R(r,\sigma,f)),
> $$
>
> where
>
> $$
>  R(r(\mathcal E^*), \sigma(\mathcal E^*), f) = \mathbb{E}[(\frac{1}{\frac{r^2\mathbb{E}[f^2]}{\sigma^2}+1})^2]\mathbb{E}[f^2]+ \mathbb{E}_{\mathcal{E}^*}[(\frac{1}{\frac{r}{\sigma}+\frac{\sigma}{r\mathbb{E}[f^2]}})^2]
> $$
>
> and
>
> $$
> R(r(\mathcal E^*), \sigma(\mathcal E^*), f)<\mathbb{E}[f^2] (\frac{1}{(L+1)^2}+ \frac{1}{L+2 + \frac{1}{L}})
> $$
>
> Here we make some remarks:
>
> * The assumption $\mathbb{E}[f(S)S]=0$ assumes the model misspecification.
> * The assumption $\frac{r^2\mathbb{E}[f^2]}{\sigma^2}>L>1$ requires that the spurious correlation between $V$ and $Y$ should be strong so that $V$ could provide more additional information for predicting $Y$. Actually, $\frac{r^2\mathbb{E}[f^2]}{\sigma^2}$ is like the signal/noise, and if it is too small, the correlation between $V$ and $Y$ in each sub-population is quite weak. And in this Theorem, we mainly focus on the **strong spurious correlation setting**, where $\frac{r^2\mathbb{E}[f^2]}{\sigma^2}$ should be large. Therefore, we use one parameter $L>1$ to characterize the strength of the spurious correlation.
> * The error bound will not grow too large or go to infinite. From $R$, we can see that the larger $L$ leads to a smaller $R$ and a tighter approximation error bound. Also, we could calculate the approximation error rate when the approximation error is worst :
> $$
> Error\ Rate \leq \max(\frac{\sigma_Y^2}{\mathcal{H}_{\mathcal V}^{\mathcal C}}, (\frac{1}{(L+1)^2}+\frac{1}{L+2+\frac{1}{L}})(\frac{L+1}{L+\text{Var}(r)}))
> $$
> * Therefore, the results show that it can be controlled even in some bad cases. And when $L$ is large (stronger spurious correlations), our approximation becomes more precise. It is encouraging since strong spurious correlations usually hurt a lot w.r.t. the out-of-distribution generalization or fairness.
>
> ### **Q2: Graph Data**
> In the second paragraph of the introduction, we review the literature and list the different meanings of heterogeneity in many fields.
> As for the graph data, a heterogeneous graph means various types of nodes and edges.
> Therefore, the heterogeneity in these works mainly refers to different data types or structures and is not in line with our work.
> In this work, from the perspective of prediction power, we focus on the data heterogeneity that affects the model prediction, which is a new type of data heterogeneity.
> In order to avoid misunderstanding, we add more explanations in the introduction to clarify the differences.

---

> > ### Author Response · Authors · 2022-11-09
> > **Q3: Regularizer $U_{\mathcal V}(W,Y_N)$**
> >
> > The regularizer $U_{\mathcal V}(W,Y_N)$ depends on the choice of the predictive function family $\mathcal V$, which we have specified in Section 4 for the regression and classification tasks (see (1) and (2) in Section 4).
> >
> > For the regression task, we typically choose $\mathcal V_1 =$ { $g:g[x]=\mathcal{N}(f(x), \sigma^2)$}, and the regularizer for this function family is:
> > $$
> > U_{\mathcal V_1}(W,Y_N) = \sum_{j=1}^K(\sum_{i=1}^Nw_{ij}y_i)^2\frac{1}{N\sum_{i=1}^Nw_{ij}} - (\frac{1}{N}\sum_{i=1}^Ny_i)^2,
> > $$
> > which is the variance of label $Y$ at the sub-population level.
> >
> > For the classification task, the function family is typically chosen as $\mathcal V_2=$ { $g:g[x]=f_\theta(x)\in \Delta_c$}, where $c$ is the number of classes and $\Delta_c$ denotes the $c$-dimensional simplex.
> > And the regularizer for this function family is:
> > $$
> > U_{\mathcal V_2}(W,Y_N) = -\sum_{j=1}^K \frac{1}{N}(\sum_{i=1}^Nw_{ij})H(Y_N^j),
> > $$
> > where $H(Y_N^j)$ denotes the entropy of $Y$ given $\mathcal E=e_j$.
> >
> > ### **Q4: $\mathcal H(P)$**
> > Here is a typo and $\mathcal{H}(P)$ refers to $\mathcal{H}^{\mathscr{E}}(P)$ in Definition 3.
> > The 'correlation' here refers to the predictive power of $X$ for predicting $Y$, and we correct this in the main body to avoid misunderstandings.
> >
> > ### **Q5: Theorem 4**
> > Theorem 4 demonstrates the exact equivalence between the empirical prediction heterogeneity and the objective function of Equation 15, and does not consider the approximation in Equations 20 and 21 during the optimization.
> > As for the bias from the approximation, theoretical results from truncated back-propagation [1] could help to analyze.
> >
> >
> > **Firstly**, in order to effectively calculate the empirical predictive heterogeneity, we formulate it as a bi-level optimization problem, and Theorem 4 is to demonstrate the exact equivalence between the empirical prediction heterogeneity and the objective function of our IM algorithm (Equation 15).
> > Then to empirically optimize Equation 15, we adopt the truncated back-propagation[1] and provide an empirical algorithm in practice.
> >
> > **Secondly**, from the theoretical results in truncated back-propagation[1], when the lower-level problem is locally strongly convex around $\theta^*(W)$, on-average convergence to an $\epsilon$-approximate stationary point is guaranteed by $O(\log 1/\epsilon)$-step truncated back-propagation.
> > In Equation 20, we perform $t$-steps of inner loop gradient descent, and the bias can be exponentially small in $t$ (see Proposition 3.1 and Lemma 3.2 in [1]).
> > And in practice, we find performing 1-step truncated back-propagation gives satisfying results, which is also found by [2].
> > Furthermore, **one can increase the number of steps to obtain better precision**, which is **supported by our approximation**.
> >
> > **Further**, since the key contribution of this work is the predictive heterogeneity measure as well as its properties, we do not pay much attention to improving the precision of the bi-level optimization techniques (e.g., its precision under general cases), which is also an important problem in the field of bi-level optimization but is not the focus of this paper.
> >
> >
> > [1] Shaban, A., Cheng, C. A., Hatch, N., & Boots, B. (2019, April). Truncated back-propagation for bilevel optimization. In The 22nd International Conference on Artificial Intelligence and Statistics (pp. 1723-1732). PMLR.
> >
> > [2] Zhou, X., Lin, Y., Pi, R., Zhang, W., Xu, R., Cui, P., \& Zhang, T. Model agnostic sample reweighting for out-of-distribution learning. In International Conference on Machine Learning (pp. 27203-27221). PMLR.

---

> > > ### Author Response · Authors · 2022-11-09
> > > **Q6: The choice of $K$**
> > >
> > > Thanks for your constructive suggestions and we add one experiment with multiple ground-truth sub-populations to demonstrate this.
> > >
> > > **Firstly**, the results in Appendix B show that the performances of some typical  OOD generalization methods plus our learned sub-populations are not affected much when $K$ is larger than the ground truth.
> > >
> > > **Secondly**, we add one experiment to show that (1) when the chosen $K$ is smaller than the ground truth, the performances of our methods will drop but are still better than ERM (2) when the chosen $K$ is larger, the performances are not affected much (consistent with the results in Appendix B).
> > >
> > > Experiment Setting: The input features $X=[S,T,V]\in\mathbb{R}^{10}$ consist of stable features $S\in\mathbb{R}^5$, noisy features $T\in\mathbb{R}^4$ and the spurious feature $V\in\mathbb R$:
> > > $$
> > > S\sim \mathcal{N}(2,2\mathbb I_5),\quad  T\sim \mathcal{N}(0, 2\mathbb I_4), \quad Y=\theta_S^TS + S_1S_2S_3+\mathcal{N}(0,0.5),
> > > $$
> > > and we generate the spurious feature via the:
> > > $$
> > > V = \theta_V^e Y + \mathcal{N}(0, 0.3),
> > > $$
> > > where $\theta_V^e$ varies across sub-populations and is dependent on which sub-population the data point belongs to.
> > > In training, we sample 8000 data points from $e_1$ with $\theta_V^1=3.0$, 1000 points from $e_2$ with $\theta_V^2=-1.0$, 1000 points from $e_3$ with $\theta_V^3=-2.0$ and 1000 points from $e_4$ with $\theta_V^4=-3.0$.
> > > Therefore, the ground-truth number of sub-populations is 4.
> > > In testing, we test the performances on $e_4$ with $\theta_V^4=-3.0$, which has strong distributional shifts from training data.
> > > The average MSE over 10 runs are:
> > >
> > > | Method       | $K=2$ | $K=3$ | $K=4$ | $K=5$ | $K=6$ | ERM   |
> > > |--------------|-------|-------|-------|-------|-------|-------|
> > > | Ours+Balance | 0.745 | 0.656 | 0.632 | 0.643 | 0.645 | 0.813 |
> > > | Ours+IRM     | 0.668 | 0.613 | 0.595 | 0.588 | 0.606 | 0.813 |
> > > | Ours+IGA     | 0.681 | 0.651 | 0.603 | 0.577 | 0.597 | 0.813 |
> > >
> > > From the results, we can see that when $K$ is smaller than the ground truth, increasing $K$ benefits the OOD generalization performance, and when $K$ is larger, the performances are not affected much, which is consistent with the results in Appendix B.
> > >
> > > **Thirdly**, how to choose an appropriate $K$ is an interesting problem and is also widely studied in the literature on traditional clustering algorithms.
> > > For our IM algorithm, we think there are mainly two ways to choose $K$:
> > >
> > > * According to the predictive heterogeneity index: When the chosen $K$ is smaller than the ground truth, our measure tends to increase quickly when increasing $K$; and when $K$ is larger than the ground truth, the increasing speed will slow down, which could direct people to choose an appropriate $K$.
> > > * According to the prediction model: Since our IM algorithm aims to learn sub-populations with different prediction mechanisms, one could compare the learned model parameters $\theta_1, \dots, \theta_K$ to judge whether $K$ is much larger than the ground truth, i.e., if two resultant models are quite similar, $K$ may be too large (divide one sub-population into two). For linear models, one can directly compare the coefficients. For deep models, we think one can calculate the transfer losses across sub-populations.
> > >
> > >
> > > **Then**, when our algorithm serves for downstream tasks (e.g., the out-of-distribution generalization task in Section 5.2), the models' performances are not quite sensitive to the choice of $K$s, as shown in our added experiments above and Appendix B. That is, if we set $K$ in an appropriate range, the performances on the downstream tasks can gain some benefits.
> > >
> > > ### **Q7: Why EIIL fails**
> > > **Firstly**, we plot the learned sub-populations of EIIL in Figure 4(b), and we can see that the relationship between the spurious feature $V$ and $Y$ does not vary much.
> > > As a result, EIIL cannot eliminate the effects of the spurious feature because of the low quality of learned sub-populations.
> > >
> > > **Secondly**, as for the reason why EIIL fails to generate good sub-populations, we think that it is due to its clustering mechanism which performs in the gradient space.
> > > Since the gradient is more sensitive to noises, the resultant clusters could be affected by the noises.
> > > To verify this, for each of the two sub-populations learned by EIIL, we run a linear regression and their coefficients are as follows:
> > >
> > > |        sub-population       | $S_1$ 	| $S_2$  	| $S_3$ 	| $S_4$  	| $S_5$ 	| $V$   	|
> > > |--------------|-------|-------|-------|-------|-------|-------|
> > > | Coefs $e_1$    	| 0.687 	| -1.147 	| 1.724 	| -0.572 	| 1.137 	| 0.349 	|
> > > | Coefs $e_2$    	| 0.995 	| -1.643 	| 2.556 	| -0.766 	| 1.556 	| 0.343 	|
> > >
> > >
> > >
> > >
> > > From the results, we can see that for the learned sub-population of EIIL, the discrepancy on stable features $S$ is much larger than that on the spurious feature $V$, which misleads the backbone IRM to use the spurious feature for prediction.

---

> > > > ### Comment · Reviewer_6bwF · 2022-11-17
> > > > **Some follow-up questions**
> > > >
> > > > Thanks for your answer. Most of my concerns have been well addressed. But I have some follow-up questions based on your updates on the paper.
> > > >
> > > > Q1: Theorem 2 focuses more on the strong spurious correlation setting. It might become loose under weak spurious correlation. It might be more convincing to provide some insights into how predictive heterogeneity works in the weak spurious correlation setting.
> > > >
> > > > Q2: There is a statement below theorem 2 that "The constant K characterizes the strength of the spurious correlation between V and Y. Larger K means V could provide more information for prediction." Should it be "L" here, since K is not defined in the theorem?

---

> > > > > ### Author Response · Authors · 2022-11-18
> > > > > **Further Comments**
> > > > >
> > > > > Thanks for your reply and the inspiring suggestions on making this work better!
> > > > >
> > > > > ### **Q1. Weak spurious correlation**
> > > > > **Firstly**, in practice, strong spurious correlations will greatly hurt the model's performance under distributional shifts and are the main targets of many OOD generalization methods.
> > > > > Thus, in our illustrative examples, we mainly focus on the settings where spurious correlations are somewhat strong, and show that our proposed measure could reflect this.
> > > > > Also, similar scenarios are considered in literatures on the theoretical analysis of spurious correlations.
> > > > > Sagawa et al. [1] also assume that the noise scale on the spurious features is quite small (see Theorem 1 in [1]).
> > > > > The setting of weak spurious correlation is interesting but has **limited practical value** since even models trained by ERM will not be much affected.
> > > > >
> > > > >
> > > > >
> > > > > **Secondly**, as suggested, we would like to discuss more under the weak spurious correlation setting.
> > > > > * Generally, when the spurious correlation is weak, the predictive power of the spurious feature $V$ to the misspecification term $f(S)$ is much reduced. Therefore, the additional information which could be provided will decrease.
> > > > > * Correspondingly, in our toy setting, the regression coefficient of $V$ is given as:
> > > > > $$
> > > > > h_V = \frac{\mathbb E[r^2(\mathcal E^*)]\mathbb E[f^2]}{\mathbb E[r^2(\mathcal E^*)]\mathbb E[f^2] + \mathbb E[\sigma^2(\mathcal E^*)]}.
> > > > > $$
> > > > > When the spurious correlation is weak (i.e. $L$ is small), the coefficient $h_V$ will be quite small.
> > > > > Therefore, the spurious feature itself could not provide much additional information for the prediction of $Y$.
> > > > > * **Further, when the spurious correlation is weak, the effects of the model misspecification term $f(S)$ may be reduced after the data splitting, which becomes the main source of the predictive heterogeneity**. However, the analysis of this depends on the specific form of $f(S)$, and we think it remains an open problem (there are few works that theoretically analyze model misspecifications). But it is indeed an interesting direction to be explored in the future.
> > > > >
> > > > > Also, when there is no model misspecification and no spurious correlation (weakest) meanwhile, Theorem 1 demonstrates that generally our measure $\mathcal H^{\mathscr E}(X\rightarrow Y)< \pi\sigma^2$, and experimental results in Appendix D (Figure 8 (a)) shows that our estimated measure is approaching 0 with the growing sample size.
> > > > >
> > > > > [1] Sagawa, S., Raghunathan, A., Koh, P. W., & Liang, P. (2020, November). An investigation of why overparameterization exacerbates spurious correlations. In International Conference on Machine Learning (pp. 8346-8356). PMLR.
> > > > >
> > > > > ### **Q2. $K$**
> > > > > Here is a typo and $K$ should be replaced with $L$. Thanks for pointing this and we have corrected it in our paper.

---

### Public Comment · ~Hongrui_Liu1 · 2023-03-13
**Some issues about this paper and very looking forward to the reply**

Recently I have read this paper "Measure the Predictive Heterogeneity". I enjoyed it and benefited a lot from it! But still I have some issues on the basic hypothesis and the proof. They are listed as follows and I would really appreciate it if authors could answer any questions for me.

1. About the basic hypothesis on the environment variable. In definition 3 it claimed that $\mathscr{E}$ is an environment set iff there exists $\mathcal{E}\in \mathscr{E}$ such that $X,Y\perp \mathcal{E}$. Also, it subsequently said that "condition for the environment set implies that the joint distribution of X, Y could be preserved in each environment". Both implied that $P(X,Y|\mathcal{E})$ is the same for all the environment $e_{1}$ and $e_{2}$ under the partition $\mathcal{E}$. But I note that your previous paper "Heterogeneous Risk Minimization" proposed that $\forall e\in supp(\mathcal{E})$ \ $supp(\mathcal{E}_{tr})$, the data and label distribution $P^{e}(X,Y)$ can be quite different from that of training environments. I am very concerned about this hypothesis because it is an important condition for the proof of the Nonnegativity of Predictive Heterogeneity. Is $X$ a causal variable? Or the "existed" partition is like a random partition for all $X$ and $Y$?

2. About the proof of Theorem 1 in Eq. 88. The definition in Eq.10 or in Eq.24 is
$H_{\mathcal{V}}^{\mathscr{E}_{K}}=\sup_\mathcal{E} \mathbb{I}_\mathcal{V}(X\rightarrow Y|\mathcal{E})-\mathbb{I}_\mathcal{V}(X\rightarrow Y)$, but why in Eq.88 it turns around.

3. About the proof of Theorem 2. From Eq.105 to Eq.106, it seems the square is not included. Especially, the denominator in Eq. 106 does not follow a square form.

4. About the objective function. In Eq. 24, $H_{\mathcal{V}}^{\mathscr{E}_{K}}$ includes two terms. But when it comes to the objective function, it only includes the first term $\mathbb{I}_\mathcal{V}(X\rightarrow Y|\mathcal{E})$. And the Theorem 4 only show $\mathbb{I}_\mathcal{V}(X\rightarrow Y|\mathcal{E})$ can be realized by the objective function. So where did $\mathbb{I}_\mathcal{V}(X\rightarrow Y)$ go?

If I have any misunderstandings or knowledge gaps, please tell me. And I am sincerely looking forward to your reply. Thank you!

---

> ### Author Response · Authors · 2023-03-14
> **Response**
>
> Thanks for your comment and our response follows.
>
> ### Question 1
> Thanks for your question and your mention of our previous work "Heterogeneous Risk Minimization"(abbr. HRM). The environment set $\mathscr E$ is defined as **a group of** environment partitions in Definition 3 rather than a single environment partition $\mathcal E $ as is characterized in HRM. Since you've been familiar with HRM, its "heterogeneity identification" step searches for the "true" environment partition and the "true" environment partition does satisfy the "heterogeneity assumption" as is described in your comment. But there must be a searching space, i.e. **a group of** environment partitions, where we explore for the "true" partition. Such searching space is instantialized as a $K$-value categorical variable in HRM, while this Predictive Heterogeneity considers a much more general definition of the environment searching space, which we call the environment set $\mathscr E$. A natural question arises that which is the minimal elements of an environment set. And similar to the fact that zero is the minimal element of a linear space, we perceive that the random environment partion is the minimal element of an environment set. It's easier to understand if you consider the procedure of "heterogeneity identification" in HRM. The search for the true environment partition (with EM algorithm) must start from some random initialization, which is exactly the "zero" element of our environment set in Definition 3.
>
> Still, your judgement is correct that Definition 3 does not ensure that our solved environment partion for Predictive Heterogeneity equals some "true" environment if there exists some causal structure in data as described in your comment. And that's what we've done in the theoretical analysis of Theorem 1&2. Theorem 1 correpsonds to the case when $X$ is a direct cause of $Y$, and we prove that the Predictive Heterogeneity is solved by random environment partition in such scenario, which is quite intuitive. On the other hand, Theorem 2 considers the case when $X$ constains some spurious variables, and we prove for specific choice of environment set, Predictive Heterogeneity is approxiamately solved by the "true" partion. And the environment set in Theorem 2 is actually the maximal environment set containing all categorical variables which corresponds to the searching space of HRM. It can be verified that the heterogeneity assumption of HRM can be satisfied for such environment set.
>
> ### Question 2
> Thanks for your suggestion. It's an typo in the proof but Theorem 1 is still correct. We've corrected the proof. We feel deeply sorry for your confusion.
>
> ### Question 3
> The proof is correct. Since Predictive Heterogeneity has four items, one combines Eq.93, Eq.104 & Eq.105 to derive the approximation of Predictive Heteorogeneity in Eq.106 & Eq.107. Please note that Eq.106 is an approximation result and the square in Eq.105 could be found in the approxiamation error of Eq.107.
>
> ### Question 4
> When it comes to the optimization for Eq.10, the supremum w.r.t. $\mathcal E$ is independent of $I_\mathcal V(X\rightarrow Y)$ from the definition in Eq.3. Thus, the solution of $\mathcal E$ for Eq.10 is exactly the solution to the overall obejective in section 4 where only $I_\mathcal V(X\rightarrow Y|\mathcal E)$ is reserved.
>
> Thanks and we welcome further discussion.

---

### Decision · Program_Chairs · 2023-01-20

**Decision:**

Accept: poster

**Justification For Why Not Higher Score:**

It is unclear how significant of an impact the work would have in practice. The reviewers believe the work is solid, but not groundbreaking.

**Justification For Why Not Lower Score:**

All reviewers believe the paper should be accepted. The topic is interesting and the technique could be useful in many applications. Most reviewer concerns have been addressed in the authors' response.

**Metareview: Summary, Strengths And Weaknesses:**

The paper defines a new notion of data heterogeneity and shows how it can be estimated from finite data. The authors provide theoretical analysis and empirical results showing the usefulness of their proposed notion.

All reviewers believe the paper should be accepted. The topic is interesting and the technique could be useful in many applications. The authors have resolved a number of reviewers' concerns during the discussion phase.

**Note From Pc:**

if the above contains the word "oral" or "spotlight" please see: "oral" presentation means -> notable-top-5% and "spotlight" means -> notable-top-25%. As stated in our emails, we are disassociating presentation type from AC recommendations